# Revising and Falsifying Sparse Autoencoder Feature Explanations

George Ma[1]*    Samuel Pfrommer[1]*    Somayeh Sojoudi[1]

[1]University of California, Berkeley

## Abstract

Mechanistic interpretability research seeks to reverse-engineer large language models (LLMs) by uncovering the internal representations of concepts within their activations. Sparse Autoencoders (SAEs) have emerged as a valuable tool for disentangling polysemantic neurons into more monosemantic, interpretable features. However, recent work on automatic explanation generation for these features has faced challenges: explanations tend to be overly broad and fail to take polysemanticity into consideration. This work addresses these limitations by introducing a similarity-based strategy for sourcing close negative sentences that more effectively falsify generated explanations. Additionally, we propose a structured, component-based format for feature explanations and a tree-based, iterative explanation method that refines explanations. We demonstrate that our structured format and tree-based explainer improve explanation quality, while our similarity-based evaluation strategy exposes biases in existing interpretability methods. We also analyze the evolution of feature complexity and polysemanticity across LLM layers, offering new insights into information content within LLMs' residual streams. Code is available at https://github.com/GeorgeMLP/feature-interp.

## 1 Introduction

The recent rise of highly capable large language models (LLMs) has inspired a surge of mechanistic interpretability research. One major concern of this area has involved decomposing LLM activations into a human-interpretable form [11, 26]. This is complicated by LLM's hypothesized use of *superposition* to encode a vast range of concepts into a constrained number of multi-layer perceptron (MLP) block neurons [3]. Namely, instead of MLP block neurons each corresponding to a clean concept, they become *polysemantic*, and activate for a range of unrelated inputs [6, 3].

Sparse Autoencoders (SAEs) have enjoyed widespread use as a more refined analysis tool that partially disentangles the polysemanticity of individual neurons [35, 3, 5, 16]. Inspired by the linear representation hypothesis [27], SAEs consist of an autoencoder which maps activations into a higher-dimensional latent space while minimizing both reconstruction loss and a sparsity penalty [5]. In contrast to individual neurons, features in the latent space of an SAE are generally more monosemantic and interpretable [5].

One area of LLM autointerpretability research attempts to explain language model features, such as neurons and SAE latent vector indices, using external LLMs to summarize patterns in top-activating sentences [1]. These generated explanations are often oversimplified and too broad [1, 28]. Consider for example the "not all" neuron L13N1352 discussed in Bills et al. [1]. This neuron activates on the token `all` only when immediately following `not`. However, a more broad explanation that the neuron always activates on the word `not` scores well on high-activation sentences which do not contain a falsifying counterexample of `not` in a different context. Various partial remedies to this problem

---

*Equal contribution. Correspondence to: George Ma (george_ma@berkeley.edu).

39th Conference on Neural Information Processing Systems (NeurIPS 2025).

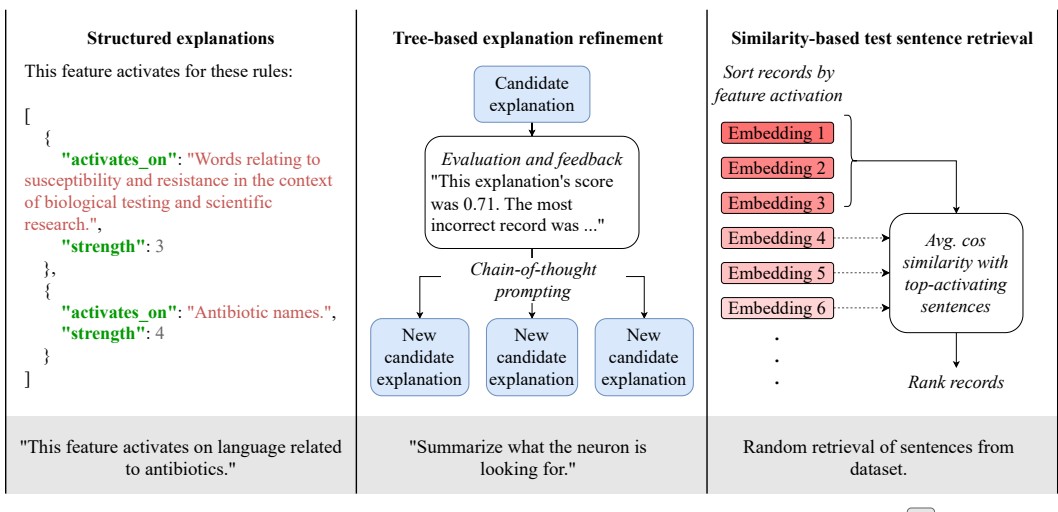

Figure 1: A high-level overview of our experiments for automatic interpretation of SAE features. Left: we represent explanations as a structured list of dictionaries. Center: we refine explanations using a tree-based optimization strategy. Right: we source challenging explanation evaluation sentences using a similarity-based scheme.

have been proposed, but they require serious compromises. For example, Bills et al. [1] generates an explanation-dependent sentences, while Paulo et al. [28] simplifies the problem to that of binary classification. Our work provides a method for sourcing semantically similar sentences from a fixed dataset for the purpose of falsifying explanations.

Explanations in previous interpretability research primarily consist of brief summary statements of patterns generated in a one-shot manner [1]. This lack of structure often obscures interesting aspects of SAE features, such as polysemanticity and the relative strengths of different feature components. Bills et al. [1] explored some strategies for explanation revision, but their unreleased implementation required generating synthetic explanation-dependent sentences. We introduce two innovations in this regard: 1) a tree-based iterative method for generating explanations, and 2) a structured representation for feature explanations which elucidates changes in SAE feature composition over the layers of an LLM.

**Contributions.** Our main contributions are listed below and summarized in Figure 1.

1. We provide a method for sourcing "close negatives" to top-activating sentences in the dataset, and show that these more effectively falsify explanations than the random sentences used in prior work.

2. We introduce a structured format for feature activations as a list of monosemantic explanations.

3. We develop a tree-based explanation method that iteratively refines its explanations through evaluation and feedback.

4. Through empirical analysis, we show that both the structured explanation format and the tree-based explainer improve the quality of feature explanations. Semantically similar negatives more effectively falsify explanations and reveal the recall bias in current interpretability methods. We further investigate how feature complexity and polysemanticity evolve across LLM layers.

## 2 Related work

**Interpreting SAE activations.** There has been significant recent research interest in explaining and simulating SAE activations. The seminal work of Bills et al. [1] introduced language model

explanation generation for MLP neurons in LLMs. Namely, the authors constructed an automated interpretability pipeline consisting of an *explainer LLM* which generated a short natural-language summary of a feature given its top-activating sentences, and a *simulator LLM* which predicts token-wise activations given the explanation. In Singh et al. [33], this idea is adapted to analyze text-to-scalar functions more generally. Recent approaches explore alternative approaches of simulating and scoring explanations by simplifying activation prediction to a classification problem [28]. Foote et al. [7] represents SAE features as a graph by identifying tokens which cause subsequent activations.

**Optimizing over strings.** A wide variety of real-world engineering problems involve iteratively optimizing over a piece of text subject to some reward signal. The LLM interpretability work of Bills et al. [1] pursued a simple prompting strategy of asking the explainer model to revise its explanation in light of novel activation data. In Pryzant et al. [30], the optimization problem was explored more generally by mimicking the gradient descent with textual prompts. LLM pipelines have also shown to benefit from a degree of automatic prompt generation and optimization [17, 18]. By using an evolutionary search strategy, the FunSearch source code generation tool was able to discover novel and efficient computing algorithms [32]. Finally, a recent Tree of Attacks with Pruning (TAP) search scheme proved highly efficient for generating LLM jailbreaking strings [21] and has been successfully adapted to other tasks such as prompt injection [29]. We use a variant of TAP for our use case due to its simplicity and effectiveness. While an extensive comparison of various optimization schemes would be valuable future research, TAP is sufficient for the purpose of this work: extracting insights about SAE features in an automated way.

**Abstractions within LLMs.** While various efforts have been made to analyze the abstraction of LLM concepts as a function of layer, this has been largely underexplored in an SAE context. One early work on the abstraction of features within transformers found that deeper layers within BERT recovered progressively more complex features in a classical NLP pipeline [36]. Another leveraged human annotations of texts which trigger attention keys and found that deeper layers generally feature more semanticity than early layers. In Jin et al. [15], linear probing on LLM activations at various layers demonstrate that certain question-answering skills are localized to particular depths. Minegishi et al. [24] finds the SAEs on deeper LLM layers are better able to disentangle the various meanings of polysemous words.

## 3 Problem formulation

Section 3.1 introduces the notation and definitions used in this paper, and Section 3.2 formalizes the problem of learning explanation-conditioned features.

### 3.1 Notation

We denote the discrete space of tokens by $\mathcal{T}$ and the associated continuous embedding space by $\mathcal{E} = \mathbb{R}^h$, where $h$ is the transformer hidden dimension. We denote the concatenation of vectors $\mathbf{x}$ and $\mathbf{y}$ using square brackets: $[\mathbf{x}, \mathbf{y}]$. The all-one vector of length $n$ is denoted by $\mathbf{1}_n$. For a vector $\mathbf{x} = [x_1, x_2, \ldots, x_n]$ and integer $i \in \{1, \ldots, n\}$, we let $\mathbf{x}_{-i} \in \mathbb{R}^{n-1}$ denote the vector produced by dropping the $i$th element.

### 3.2 Feature learning

Let a *feature* be a function $f \colon \mathcal{T}^n \to \mathbb{R}_+^n$ which maps a token sequence $\mathbf{t} \in \mathcal{T}^n$ to a vector of *feature expressions* $f(\mathbf{t}) \in \mathbb{R}_+^n$. The space of all features is denoted by $\mathcal{F}$.

Consider a particular ground-truth feature $f^* \in \mathcal{F}$, perhaps consisting of the activations of an SAE latent index. We can associate $f^*$ with a dataset of associated feature expressions

$$\mathcal{D}_{f^*} = \left\{ \left( \mathbf{t}_i, f^*(\mathbf{t}_i) \right) \right\}_{i=1}^d .$$

Now consider the problem of learning an explanation-conditioned feature $g_\theta \colon \mathcal{T}^n \to \mathbb{R}_+^n$ which reproduces the feature patterns of $f^*$ in $\mathcal{D}_{f^*}$. We let the explanations $\theta$ fall in the space of structured natural-language descriptions $\theta \in \mathcal{T}_{\mathrm{NL}}^l$, and leverage a language model to simulate the predictions of $g_\theta$ for a particular token sequence $\mathbf{t} \in \mathcal{T}^n$ [1]. Precisely characterizing "natural-language" text is

challenging. However, as we ultimately use language models to search over explanations, we observe that the constraint $\theta \in \mathcal{T}_{\mathrm{NL}}^l$ is satisfied in practice.

When evaluating the performance of $g_\theta$ in reproducing the activations of $f^*$, it is computationally infeasible to evaluate over all of $\mathcal{D}_{f^*}$. We thus follow the practice of Bills et al. [1] in subsampling $\mathcal{D}_{f^*}$ for evaluation.

We first define the subset $\mathcal{D}_{f^*}^+ \subset \mathcal{D}_{f^*}$ of *top-activating* sentences as a random sample of $k$ sentences from the top quantile of $\mathcal{D}_{f^*}$ ordered by $\|f^*(\mathbf{t}_i)\|_\infty$. We then define a set of *complementary* sentences $\mathcal{D}_{f^*}^{\mathbf{ss}} \subset \mathcal{D}_{f^*} \setminus \mathcal{D}_{f^*}^+$, chosen according to a sampling strategy $\mathbf{ss}$ resulting in a cardinality $|\mathcal{D}_{f^*}^{\mathbf{ss}}| = |\mathcal{D}_{f^*}|$. These sentences are intended as "negatives" that falsify an overly-broad explanation of the top activations in $\mathcal{D}_{f^*}^+$ [1]. We empirically compare sampling strategy choices in Section 5.1.

The *dataset loss* is then defined using the standard Pearson correlation coefficient metric defined in Bills et al. [1]:

$$\mathcal{L}(\theta) = \mathrm{corr}\Big( [f^*(\mathbf{t}_{i_1}), f^*(\mathbf{t}_{i_2}), \dots]_{i_j \in \mathcal{I}_{\mathcal{D}_{f^*}^+, \mathcal{D}_{f^*}^{\mathbf{ss}}}}, [g_\theta(\mathbf{t}_{i_1}), g_\theta(\mathbf{t}_{i_2}), \dots]_{i_j \in \mathcal{I}_{\mathcal{D}_{f^*}^+, \mathcal{D}_{f^*}^{\mathbf{ss}}}} \Big), \tag{1}$$

where with some abuse of notation we let $\mathcal{I}_{\mathcal{D}_{f^*}^+, \mathcal{D}_{f^*}^{\mathbf{ss}}}$ be the sequence of data indices selected in the construction of $\mathcal{D}_{f^*}^+$ and $\mathcal{D}_{f^*}^{\mathbf{ss}}$.

# 4 Method

Section 4.1 briefly summarizes the simulation method from Bills et al. [1] for completeness. In Section 4.2, we provide details regarding our explanation generation schemes. Section 4.3 motivates and defines our structured schema for feature explanations. Finally, Section 4.4 discusses how we source "close counterexamples" for top-activating sentences from our dataset.

## 4.1 Feature simulation

Consider an explanation $\theta$ for a feature, and recall that the associated feature simulator $g_\theta : \mathcal{T}^n \to \mathbb{R}_+^n$ maps a sequence of tokens to a sequence of nonnegative reals. Our simulation approach is minimally modified from Bills et al. [1]. Namely, we provide a simulating LLM the explanation and a formatted list of tokens $\mathbf{t} \in \mathcal{T}^n$, where each token is listed on a new line followed by a `tab` and an `unknown` token. We do one inference pass per sentence and compute the expected feature expression for each token by examining the negative log probabilities on the `unknown` tokens. Unfortunately, the original implementation in Bills et al. [1] relied on OpenAI API calls returning the top-$k$ log probabilities for tokens in the input prompt. This has been deprecated and to the best of our knowledge is not available in other cloud providers. We thus perform simulation locally using the `gemma-2-27b-it` model with a bilevel key-value caching scheme for efficiency (Appendix A.2). For simulating structured explanations, we perform multiple passes of the above simulation scheme, one per explanation component. Further details are deferred to Section 4.3.

## 4.2 Explanation generation

We outline two different ways of generating explanations: a one-shot method from Bills et al. [1], and a novel tree-based method inspired by Mehrotra et al. [21]. We also experiment with including *holistic activations* to improve explanation quality.

**One-shot explanation generation.** The one-shot explainer includes a system prompt which explains the explanation schema and the activation record format [1]. This is then followed by a series of few-shot examples, each containing a sequence of activation records as a user chat message followed by the associated structured explanation in the assistant response. The final message to the model consists of top-activating records $\mathcal{D}_{f^*}^+$ for the feature of interest. Each activation record is formatted as in Bills et al. [1], with activations for each token normalized and rounded to be integers between $0$ and $5$.[2]

---

[2]Bills et al. [1] formatted activations to be between $0$ and $10$. We found that some tokenizers subdivided $10$ into two tokens, and we thus adjusted the range maximum to $5$.

**Tree-based explanation generation.** We now introduce a novel explanation procedure based on the Tree of Attacks with Pruning (TAP) jailbreaking technique [21]. We initialize with $w \in \mathbb{N}$ root leaf nodes generated via one-shot explanation, and iteratively perform the following steps $d \in \mathbb{N}$ times:

1. *Evaluation and feedback.* Simulate the activations for each explanation and score them against the ground-truth activations via (1). Construct a feedback message consisting of the correlation score as well as the correct and simulated feature expressions for the lowest-performing sentence in the training data.

2. *Branching.* For each leaf node, generate $b$ child explanations via chain-of-thought reasoning, providing the feedback message as well as entire conversation history for the ancestors of the leaf node.

3. *Pruning.* Retain the $w$ leaf nodes with the highest training dataset score for the next layer of the tree.

The algorithm is terminated after $d$ iterations or after a certain score threshold is met. All the explanations produced at each layer of the tree are then scored using a validation dataset, and the best-performing explanation is returned. The pseudocode of the tree-based explainer is shown in Algorithm 1.

---

**Algorithm 1** Tree-Based Explainer for SAE Feature Interpretation

---

**Require:** Training and validation records for a single SAE feature, each containing top-activating and complementary sentences with ground-truth activations
**Ensure:** Natural language explanation for the target SAE feature
 1: **Initialize:** Prompt the one-shot explainer (as in Bills et al. [1]) $w$ times to generate $w$ initial explanations as leaf nodes
 2: **for** $i = 1, 2, \ldots, d$ **do**
> ▷ — Evaluate current leaf nodes —
 3:    **for** each leaf node explanation $\theta_j$ **do**
 4:        Provide $\theta_j$ and validation sentences to a simulator LLM
 5:        Obtain simulated activations $\hat{\mathbf{a}}_j$ from the simulator
 6:        Compute correlation score $s_j = \mathrm{corr}(\hat{\mathbf{a}}_j, \mathbf{a}_j^{\mathrm{true}})$
 7:        Construct feedback message $F_j$ including:
        (a) correlation score $s_j$
        (b) simulated activations $\hat{\mathbf{a}}_j$
        (c) ground-truth activations $\mathbf{a}_j^{\mathrm{true}}$ for the lowest-performing sentence
 8:    **end for**
> ▷ — Generate improved explanations —
 9:    **for** each leaf node explanation $\theta_j$ **do**
10:        Provide $F_j$ and conversation history of ancestor nodes to the LLM
11:        Use chain-of-thought prompting to generate $b$ improved explanations
12:        Add the new explanations as child nodes in the tree
13:    **end for**
> ▷ — Evaluate and select new leaf nodes —
14:    **for** each new leaf node explanation **do**
15:        Simulate and score as above to obtain $s_j$
16:    **end for**
17:    Retain the top $w$ leaf nodes with highest validation scores
18:    **if** highest validation score $> \tau$ **then**
19:        **break**
20:    **end if**
21: **end for**
22: **return** The explanation with the highest validation score across all iterations

---

**Holistic features.** We introduce holistic features as a supplementary signal which indicates what tokens causally influence SAE activations. Consider an arbitrary but particular feature $f : \mathcal{T}^n \to \mathbb{R}_+^n$.

```
[
    {
        "activates_on": string
        "strength": [0, 1, 2, 3, 4, 5]
    },
    ...
]
```

Figure 2: The schema for our structured explanations.

We define the associated *holistic feature* $\tilde{f} : \mathcal{T}^n \to \mathbb{R}^n$ as:

$$\tilde{f}(\mathbf{t})_i = \frac{1}{n} \left( \mathbf{1}_n^\top f(\mathbf{t}) - \mathbf{1}_{n-1}^\top f(\mathbf{t}_{-i}) \right).$$

Holistic features measure how much a particular token impacts the total expression of a feature across all indices. Consider a feature which activates on the token `want` but only when immediately following `don't`. The associated holistic feature would be positive for both tokens, as dropping either of them from the sequence reduces the summed expression of the original feature. In our work, we experiment with supplementing feature records with holistic activations. Some examples of holistic activations are shown in Appendix B.

### 4.3 Structured explanations

Feature explanations from previous works typically are limited to short strings. For example, the explanation for L30N2902 from Bills et al. [1] is "words related to the concept of being a significant or integral part of something." However, a closer examination reveals that this explanation fails to capture activations on words relating to depth or recesses. This suggests that short, unstructured explanations obscure the polysemantic attributes of a feature.

We address this with our *structured explanation* scheme, summarized in Figure 2. An explanation consists of a list of JSON objects which we call *explanation components*. Each component contains two fields: a string description of the activations and an integer attribute for the strength of the activation between $0$ and $5$. We prompt our explainer LLM such that each component corresponds to a monosemantic concept; intuitively, a feature that requires many explanation components to achieve good performance is polysemantic.

When simulating a structured explanation, we predict activations for each component individually using the `activates_on` field and the approach in Section 4.1. We then scale each component's activations proportionally to the `strength` field and combine by taking the maximum activation across components for each token. This ensures that the simulated feature activations are invariant to permutations of the structured explanation list.

### 4.4 Complementary sentence sourcing

We propose four methods for selecting complementary sentences $\mathcal{D}_{f^*}^{\text{ss}}$ in the evaluation dataset: (1) the `random` strategy, where sentences are randomly sampled from the entire dataset $\mathcal{D}_{f^*} \setminus \mathcal{D}_{f^*}^+$; (2) the `similar` strategy, where sentences with the highest semantic similarity to the top-activating sentences $\mathcal{D}_{f^*}^+$ are selected; and (3) non-activating variants of these two strategies, where only sentences with no ground-truth feature activation (i.e., $\max_i f^*(\mathbf{t})_i = 0$) are considered.

**Random sentences.** Following Bills et al. [1], we randomly sample an equal number of sentences to the top-activating sentences from the entire dataset. The combined set of top-activating and randomly selected sentences is then used to evaluate feature explanations.

**Random non-activating sentences.** In this strategy, we randomly sample sentences from the dataset that have no ground-truth feature activation. This allows us to assess the ability of different complementary sentence selection strategies in finding "close counterexamples" by analyzing their false positive rate—defined as the proportion of positive elements in the explanation-conditioned feature activation $g_\theta(\mathbf{t})$, where $\mathbf{t} \in \mathcal{D}_{f^*}^{\text{ss}}$.

Current feature explanation methods are often biased toward recall: a broad explanation can activate on many relevant tokens while still achieving a good evaluation score [4]. By sourcing "close counterexamples" to the top-activating sentences, we can better assess the precision of the explanations.

**Similarity-based sentences.** Instead of randomly selecting from the entire dataset, we sample sentences with the highest semantic similarity to the top-activating sentences. Sentence similarity is measured using cosine similarity between sentence embeddings produced by a pre-trained Sentence Transformer [31]. This method involves two steps:

1. Compute the embeddings of all sentences in the dataset using a pre-trained Sentence Transformer.

2. For each SAE feature, we identify a fixed set of top-activating sentences (400 in our experiments). We then select the sentences with the highest average cosine similarity to this set and designate them as the complementary sentences for evaluation. These complementary sentences are pre-computed and cached prior to experimentation.

As we demonstrate in Section 5.1, evaluating feature explanations on these semantically similar sentences reveals a substantially higher number of false positives during the feature simulation step (Section 4.1). Specifically, the feature simulator $g_\theta$ often predicts positive activations $g_\theta(\mathbf{t}_i) > 1$ where the ground truth activation $f^*(\mathbf{t}_i)$ is zero. Since current auto-interpretability approaches are heavily biased toward recall, evaluation against these similarity-based complementary sentences offers a more reliable estimate of explanation precision, helping to expose overly broad or imprecise feature descriptions.

**Similarity-based non-activating sentences.** This method follows the similarity-based strategy but filters for sentences with no ground-truth feature activation. This ensures a fair comparison between the false positive rates of similarity-based and random selection methods. As shown in Section 5.1, similarity-based complementary sentences tend to yield higher false positive rates, leading to lower correlation scores for feature explanations. This result highlights the recall bias in current feature explanation methods and provides a more precise measure of their precision.

## 5 Experiments

In Section 5.1, we compare various methods for sourcing complementary sentences. Section 5.2 documents our improvements to the explanation generation process. Finally, Section 5.3 analyzes the impact of our structured explanations on the composition of SAE features as a function of layer.

**Common setup.** All experiments are conducted using an uncopyrighted subset of the Pile [8, 25]. We conduct experiments on a subset of 100,000 sentences and chunk them into sequences with 32 tokens. For all experiments, we use the open-source Llama 4 Scout to generate explanations [22]. Our subject language models are `gemma-2-9b`, `llama-3.1-8b`, and `gpt-2-small`. We leveraged pretrained SAEs of comparable widths, using the 16k Gemma scope SAEs [19] and 32k Llama scope and GPT-2 SAEs [9, 13]. We use the first 50 SAE features of each layer in our experiments.

### 5.1 Complementary sentence sourcing

In this section, we evaluate the four complementary sentence sourcing strategies introduced in Section 4.4. Following the setup of Bills et al. [1], we provide the explainer LLM with 10 top-activating sentences as the training dataset for each feature and prompt it to generate an explanation. We then use a simulator LLM to predict feature activations on a test dataset, which consists of 10 top-activating sentences and 10 complementary sentences. Finally, we assess the quality of the explanations by measuring (1) the false positive rate of simulated activations on the complementary sentences and (2) the correlation between ground-truth and simulated activations. We experiment with `gemma-2-9b` as our subject model.

Figure 3 shows the number of false positives per sentence of simulated feature activations on complementary sentences using structured explanations. For a fair comparison, we highlight all non-activating complementary sentence sources, as they contain no ground-truth activations (i.e., no

true positives or false negatives). We also plot the complementary sources without the no-activation restriction in dashed lines; however, comparisons involving these methods are not entirely fair since they differ in the number of true positives as well.

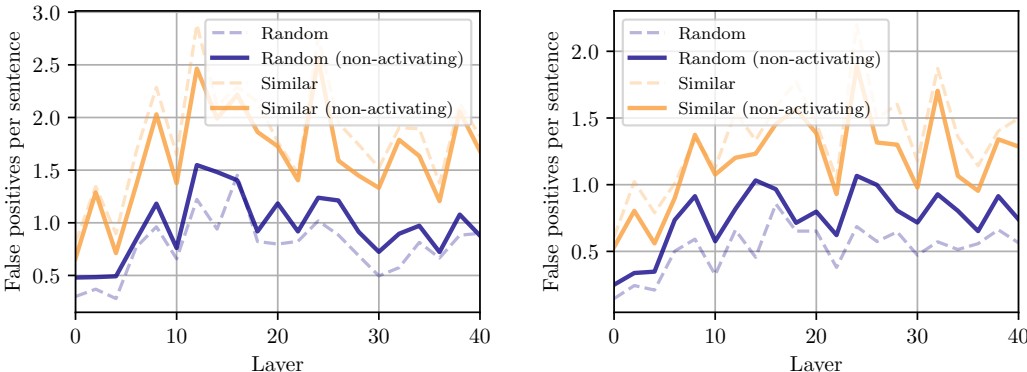

Figure 3: False positives per sentence of simulated activations on complementary sentences for the one-shot explainer (**left**) and the tree-based explainer (**right**), averaged over 50 features per layer. "Non-activating" indicates the sentences have no ground-truth activation.

As observed in Figure 3, for both the one-shot and tree-based explainer, similarity-based complementary sentence sourcing methods result in higher false positive rates than random selection methods. This indicates that similarity-based methods are more effective at identifying "close counterexamples" that expose the limitations of feature explanations in terms of precision. As noted by Caden Juang et al. [4], existing feature explanation generation methods tend to be biased toward recall. By evaluating explanations on similarity-based complementary sentences, we achieve a more precise assessment of their accuracy.

We show some examples of similarity-based complementary sentences and their corresponding simulated activations in Appendix C.

## 5.2 Explanation generation

An overview of our explanation generation results is available in Figure 4, with the associated tabular results in Table 1. As in Section 5.1, we use 10 top-activating sentences as the training set for generating the explanations, with the optional inclusion of 10 additional complementary sentences as training negatives. For the evaluation dataset, we use 10 top-activating sentences and 10 complementary sentences. We take similarity-based non-activating sentences as the complementary set.

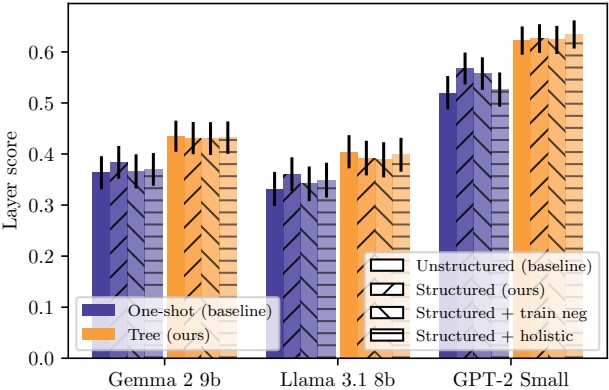

Figure 4: Explanation strategy comparison. Bars denote 90% confidence intervals.

We note that tree-based generation consistently outperforms one-shot generation.

For the unstructured explanation case, we see relative improvements of 19.8% for Gemma, 21.6% for Llama, and 16.4% for GPT-2 when using the tree explainer. In the one-shot explainer case, we also see an improvement from structured explanations, with relative improvements of 5.8% for Gemma, 8.4% for Llama, and 9.2% for GPT-2. When employing tree-based generation, structured explanations no longer provide a benefit. We hypothesize that this is attributable to tree-based generation's ability to iteratively pack polysemantic meanings into a single explanation string.

We report negative results for both the inclusion of holistic activations and training negatives in the explanation process. Figure 4 shows that including holistic activations and training negatives does not improve the performance for either the one-shot or tree explainer in any of our tested language models. Taken together, these results suggest that LLMs still struggle to generate feature explanations which account for prior context.

### 5.3 Analysis of structured explanations

We analyze feature complexity and polysemanticity for eight evenly-spaced layers within each model, including the first and final layers.

**Feature complexity.** We use an LLM judge to evaluate the complexity of generated structured explanations from the tree explainer [38]. We provide the judge with few-shot examples of explanations and the associated complexity score, ranging from 0 to 5. Lower complexity scores correspond to simple token-specific features (e.g., "the word 'instruments'") and higher complexity scores are assigned to more abstract features (e.g., "expressions of skepticism"). Our complexity judge implementation follows the simulation approach detailed in Section 4.1 and uses the `gemma-2-27b-it` model. We compute the simple average of the complexity scores across all explanation components for a particular feature.

**Feature polysemanticity.** For each feature, we run the tree explainer with rule caps ranging from 1 to 5. Each generated explanation is scored against the evaluation sentences, and we record the smallest rule cap for which 90% of the maximum score is achieved over all five settings. This intuitively provides a measure of feature polysemanticity.

**Results.** Figure 5 shows the average explanation complexity and polysemanticity of structured explanations generated by the tree explainer. We find that the Gemma 2 and Llama 3.1 models peak in complexity at the middle layers, while GPT-2's complexity remains relatively stable across layers. Gemma 2 and Llama 3.1 also exhibit a similar increasing pattern in the polysemanticity figure, rising from an average polysemanticity of 1.6 and 1.5 at the first layer, to 2.0 and 2.1 at the final layer. In contrast, GPT-2 has consistently less polysemantic features.

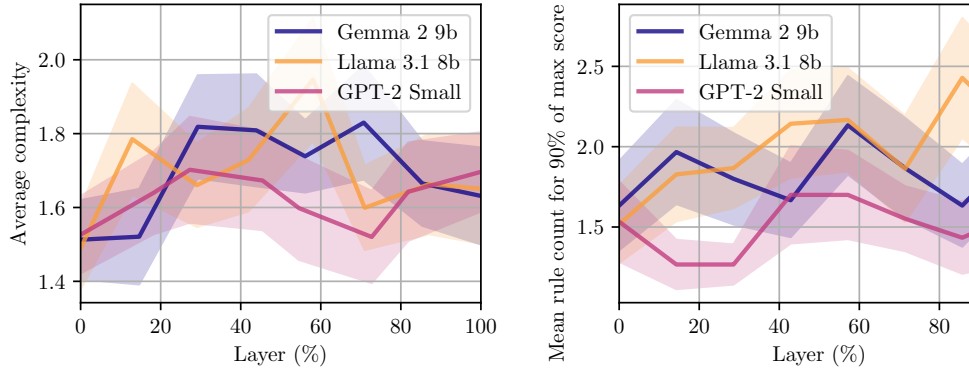

Figure 5: Lines denote mean values, and shaded regions denote 80% confidence intervals. The horizontal axis corresponds to the analyzed layer as a proportion of the total layer count, which varies between models. **Left.** The average explanation complexity for the tree explainer. **Right.** The polysemanticity of structured explanations generated by the tree explainer.

# 6 Limitations

While our methods improve the precision and falsifiability of SAE feature explanations, several limitations remain. First, the iterative tree-based explanation generation process is computationally expensive, taking approximately 1.5 minutes per feature on our hardware, which presents challenges for large-scale analysis.[3] Second, our approach relies primarily on top-activating records, which may overlook certain forms of polysemanticity—particularly when important activation patterns occur in less frequent or lower-ranked contexts. Third, as with all SAE-based interpretability methods, the resulting feature decomposition reflects the representational structure of activations rather than the true underlying computational mechanisms of the model. Consequently, while improved explanations enhance the interpretability of individual features, they do not in themselves guarantee a full understanding of model reasoning or behavior. Finally, our analyses in Section 5 focus on layer-wise and model-size trends that provide empirical insights into representational structure, but they should be viewed as preliminary steps toward mechanistic understanding rather than conclusive explanations. We hope that future work will build on these results to establish stronger connections between feature-level interpretability and higher-level model behaviors.

# 7 Conclusion

This work addresses key challenges in the automatic interpretability of Sparse Autoencoder (SAE) features in large language models. We introduce a method for sourcing semantically similar counterexamples that more effectively falsify overly broad explanations, a structured format for representing feature activations, and a tree-based iterative explainer for refining explanations through evaluation and feedback. Our experiments show that these approaches improve explanation precision, reveal the recall bias in existing methods, and provide new insights into how feature complexity and polysemanticity evolve across model layers. Together, these contributions offer a more rigorous and falsifiable framework for LLM interpretability.

## Acknowledgement

This research was supported by the U.S. Army Research Laboratory and the U.S. Army Research Office under Grant W911NF2010219, the Office of Naval Research, and the National Science Foundation. This work used Jetstream2 [12, 2] at Indiana University through allocation CIS240843 from the Advanced Cyberinfrastructure Coordination Ecosystem: Services & Support (ACCESS) program, which is supported by National Science Foundation grants #2138259, #2138286, #2138307, #2137603, and #2138296.

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

# A  Explanation, simulation, and complexity analysis

This section documents the methodological details regarding our explanation, simulation, and complexity analysis.

## A.1  Explanation

**Hyperparameters.** For the one-shot explainer, we use a temperature of 1 and a top-p value of 1 for explanation generation, with a maximum of 10 rules (i.e., components in the structured explanation). For the tree-based explainer, we set the temperature to 1.2 and top-p to 1, with a maximum of 5 rules. The tree is initialized with 3 root nodes, a maximum depth of 2, and a branching factor of 2, meaning each node generates 2 candidate explanations after evaluation and feedback. The width is set to 2, retaining the top 2 scoring explanations at each iteration and discarding the rest.

**Prompts.** For both the unstructured and structured explainers, the explanation prompt consists of three components: (1) a system prompt (provided as system messages), (2) few-shot examples (provided as user messages), and (3) their corresponding explanations (provided as assistant messages). The system prompt for the unstructured explainer is shown in Figure 6.

```
We're studying neurons in a neural network. Each neuron looks for some
↪  particular thing in a short document. Look at the parts of the
↪  document the neuron activates for and summarize in a single sentence
↪  what the neuron is looking for. Don't list examples of words.

The activation format is token<tab>activation. Activation values range
↪  from 0 to 5. A neuron finding what it's looking for is represented by
↪  a non-zero activation value. The higher the activation value, the
↪  stronger the match.

Activation records consist of two parts: activating tokens and
↪  activation-causing tokens. Activating tokens are the tokens that the
↪  feature activates on. Activation-causing tokens are the tokens that
↪  cause the feature to activate on later activating tokens.
```

Figure 6: The system prompt for the unstructured explainer. The texts in blue are optional and only included when using the holistic activations introduced in Section 4.2.

In Figure 6, the text highlighted in blue is included only when incorporating holistic activations, as described in Section 4.2. Similarly, the system prompt for the structured explainer is shown in Figure 7, where the blue text indicates content specific to holistic activations. Additionally, the purple text in Figure 7 appears only when explicitly setting a maximum number of explanation components for the structured explainer.

```
We're studying features in a neural network. Each feature looks for
↪   specific patterns in text. Analyze the parts of the text where the
↪   feature activates and explain its behavior in a structured format.

For each feature, provide a list of rules, where each rule consists of two
↪   fields:

1. 'activates_on' (string): The specific tokens on which the activation
↪   occurs. This must be a string -- NOT a list of strings.
2. 'strength' (int): The strength of the activation, from 0 to 5. Only put
↪   a single integer here, no additional text.

Each rule should consist of a single human-interpretable concept. Do not
↪   try to pack completely unrelated activating tokens into the same rule.
↪   For example, if the feature activates on the word 'stop' and also on
↪   the word 'cookie', you should put them in different rules.

But sufficiently similar or conceptually related activating tokens can be
↪   grouped together in the same rule. For example, if the feature
↪   activates on the word 'car' and also on the word 'truck', you should
↪   put them in the same rule.

The activation format is token<tab>activation. Values range from 0 to 5.
↪   Non-zero activations indicate the feature found what it's looking for.
↪   Higher values indicate stronger matches.
```



```
Activation records consist of two parts: activating tokens and
↪   activation-causing tokens. Activating tokens are the tokens that the
↪   feature activates on. Activation-causing tokens are the tokens that
↪   cause the feature to activate on later activating tokens.
```



```
Try to keep the 'activates_on' field short. Also keep the list of rules as
↪   short as possible. Only add rules to the list if they are really
↪   necessary; i.e., only add a rule if the feature activates on it and
↪   it's not already in the list.
```



```
The strict maximum number of rules is {rule_cap}. Do not generate more
↪   than this number of rules. You should not try to fill up the rule cap,
↪   only add rules if they are actually necessary and try to keep the list
↪   of rules as short as possible.
```



```
Format your response as a JSON list of dictionaries with 'activates_on'
↪   and 'strength' fields.
```

Figure 7: The system prompt for the structured explainer. The texts in blue are optional and only included when using the holistic activations introduced in Section 4.2. The texts in purple are only included when the maximum number of rules rule_cap is explicitly set.

```
RECORD START                      <start>
Activating tokens                 {tokens[0]}    unknown
<start>                           {tokens[1]}    unknown
{tokens[0]}    unknown           ...
{tokens[1]}    unknown           {tokens[n]}    unknown
...                              <end>
{tokens[i]}    {activations[i]}
{tokens[i+1]}    {activations[i+1]}
...
{tokens[n]}    {activations[n]}
<end>

Activation-causing tokens
<start>
{tokens[0]}    unknown
{tokens[1]}    unknown
...
{tokens[i]}
↪  {holistic_activations[i]}
{tokens[i+1]}
↪  {holistic_activations[i+1]}
...
{tokens[n]}
↪  {holistic_activations[n]}
<end>
RECORD END
```

(a)                                             (b)

Figure 8: Formatting of records for simulation and explanation generation. Figure 8a shows the formatting of a few-shot example record. For explanation, all few-shot examples' activations are fully shown. For simulation, the first $i$ tokens' activations are masked with `unknown` (as in Bills et al. [1]), where $i$ is an integer between $0$ and $n$. The few-shot example formatting in Figure 8a, except without any activation masking. The texts in blue are optional and only included when using the holistic activations introduced in Section 4.2. The format for the sequence being simulated is show in Figure 8b; all tokens are masked as `unknown` and the log probabilities for each score integer $0, 1, \ldots, 5$ is used to compute the expected score.

Next, we provide the few-shot examples along with their corresponding unstructured and structured explanations. These records are ultimately formatted as in Figure 8 when included in the context—however, here we highlight activations in a more human-readable way. In these examples, both regular activations and holistic activations are highlighted, with darker background colors indicating higher activation values. For readability, some line breaks from the original records have been omitted. The same set of examples is used to prompt all explainers.

## Example 1

*Regular activations*:
- javascript to provide you with a positive online shopping experience. To enable javascript
- and 3 and Are Negative for Claudin 4. Invasive apo

*Holistic activations*:
- javascript to provide you with a positive online shopping experience. To enable javascript
- and 3 and Are Negative for Claudin 4. Invasive apo

*Unstructured explanation*: `the words 'positive' and 'negative'`

*Structured explanation*:

```
[
    {
        "activates_on": "The words 'positive' and 'negative', in upper
        ↪  and lowercase.",
        "strength": 3
    },
]
```

## Example 2

*Regular activations*:
- <title>Installation of Less</title> <para>Install Less
- <title>CodeMirror: HTML mixed mode</title> <meta

*Holistic activations*:
- <title>Installation of Less</title> <para>Install Less
- <title>CodeMirror: HTML mixed mode</title> <meta

*Unstructured explanation*: `the HTML tag component '</' following an opening '<title' tag`

*Structured explanation*:

```
[
    {
        "activates_on": "The HTML tag component '</' when following an
        ↪  opening '<title' tag.",
        "strength": 4
    },
]
```

**Example 3**

*Regular activations*:

- third-generation cephalosporin-resistant Escherichia coli from broilers, swine,
- NCLLS standard for susceptibility testing of yeasts?]. The Etest and

*Holistic activations*:

- third-generation cephalosporin-resistant Escherichia coli from broilers, swine,
- NCLLS standard for susceptibility testing of yeasts?]. The Etest and

*Unstructured explanation*: `language related to biological resistance`

*Structured explanation*:

```
[
    {
        "activates_on": "Words relating to susceptibility and
        ↪  resistance in the context of biological testing and
        ↪  scientific research.",
        "strength": 3
    },
    {
        "activates_on": "Antibiotic names.",
        "strength": 4
    }
]
```

**Example 4**

*Regular activations*:

- hage migration inhibitory factor in allergic rhinitis: its identification in eosinophilsat
- prevalence of asthma and allergic disorders was assessed in 9-11 year-

*Holistic activations*:

- hage migration inhibitory factor in allergic rhinitis: its identification in eosinophilsat
- prevalence of asthma and allergic disorders was assessed in 9-11 year-

*Unstructured explanation*: `language related to diseases, disorders and allergies`

*Structured explanation*:

```
[
    {
        "activates_on": "The word 'allergic'.",
        "strength": 3
    },
    {
        "activates_on": "Diseases and disorders, only if following the
        ↪  word 'allergic'.",
        "strength": 4
    }
]
```

**Example 5**

*Regular activations*:

- ○ communication system in which Av apparatus, such as a video tape recorder (VTR
- ○ camera view at the same time? I have a requirement to show both rear

*Holistic activations*:

- ○ communication system in which Av apparatus, such as a video tape recorder (VTR
- ○ camera view at the same time? I have a requirement to show both rear

*Unstructured explanation*: `language related to views, videos, and recording`

*Structured explanation*:

```
[
    {
        "activates_on": "Words generally related to views, videos, and
        ↪  recording.",
        "strength": 3
    }
]
```

Following these few-shot examples, the explainer receives feature activation records and is prompted to generate an explanation based on them.

For the tree explainer, each iteration involves evaluating the explanations generated at each tree node, providing feedback, and prompting the node to refine its explanation. To achieve this, we first assess the explanation using validation records and identify the record with the lowest score, which is then included in the feedback prompt. Both the ground-truth activations and the activations predicted by the simulator are provided in the feedback (Figure 9).

Subsequently, we append a suffix to the feedback, instructing the explainer to reflect on the errors in its previous explanation and generate a revised one. For the unstructured tree explainer, the feedback suffix is shown in Figure 10, where the purple text is included only when explicitly setting a maximum number of rules in the structured explanations. For the structured tree explainer, the feedback suffix is shown in Figure 11.

```
Overall score (out of 1.0 max): {score}
Most incorrect record:

Activation-causing tokens (review)
<start>
{tokens[0]}    unknown
{tokens[1]}    unknown
...
{tokens[i]}    {holistic_activations[i]}
{tokens[i+1]}   {holistic_activations[i+1]}
...
{tokens[n]}    {holistic_activations[n]}
<end>

Activating token errors
<start>
{tokens[0]} (predicted: {predicted_activations[0]}) (actual:
↪  {activations[0]}) (error: {predicted_activations[0] - activations[0]})
{tokens[1]} (predicted: {predicted_activations[1]}) (actual:
↪  {activations[1]}) (error: {predicted_activations[1] - activations[1]})
...
{tokens[n]} (predicted: {predicted_activations[n]}) (actual:
↪  {activations[n]}) (error: {predicted_activations[n] - activations[n]})
<end>
```

Figure 9: The feedback string for tree-based generation for both the structured and unstructured. The text in blue is optional and is only included when using the holistic activations introduced in Section 4.2.

```
Try generating a better explanation, taking into account this feedback. Be
↪  creative; if you have been trying something and it isn't working, try
↪  something else.

Format your response as including first an improvement (in natural
↪  language), then the explanation.

You must make your improvement precise and specific; here are a few
↪  examples:
- "The neuron is activating on the word 'cat', but my explanation doesn't
↪  capture this. I should amend my explanation."
- "My score is much lower than previous attempts. I should undo my recent
↪  changes."
- "My last explanation says that 'cat' activates in any context, but this
↪  examples show that it doesn't activate when 'dog' appears previously
↪  in the text. I should narrow the context of my explanation."

The strict maximum number of rules is {rule_cap}. Do not generate more
↪  than this number of rules. You should not try to fill up the rule cap,
↪  only add rules if they are actually necessary and try to keep the list
↪  of rules as short as possible.

Structure your response as follows:

IMPROVEMENT: ...
NEW EXPLANATION: ...
```

Figure 10: The feedback suffix for the unstructured tree explainer. The text in purple is only included when the maximum number of rules is explicitly set.

```
Try generating a better explanation, taking into account this feedback. Be
↪   creative; if you have been trying something and it isn't working, try
↪   something else.

Format your response as including first an improvement (in natural
↪   language), then the explanation.

You must make your improvement precise and specific; here are a few
↪   examples:
- "The feature is activating on the word 'cat', but my explanation doesn't
↪   capture this. I should add another rule to my explanation list."
- "My score is much lower than previous attempts. I should remove the
↪   rules I added recently, perhaps they are too long and confusing."
- "My last rule to says that 'cat' activates in any context, but this
↪   examples show that it doesn't activate when 'dog' appears previously
↪   in the text. I should add this required context to the rule."
- "My rule says to activate on 'cat' with a strength of 2, but the
↪   examples have higher activations. I don't need to add any new rules,
↪   but I should increase the strength of my rule."

The strict maximum number of rules is {rule_cap}. Do not generate more
↪   than this number of rules. You should not try to fill up the rule cap,
↪   only add rules if they are actually necessary and try to keep the list
↪   of rules as short as possible.

Structure your response as follows:

IMPROVEMENT: ...
NEW EXPLANATION: [ { "activates_on": ..., "strength": ... }, ... ]

Note that the "activates_on" field is a single string, not a list of
↪   strings. Do not include any comment in the JSON list.
```

Figure 11: The feedback suffix for the structured tree explainer. The text in purple is only included when the maximum number of rules is explicitly set.

## A.2 Simulation

For simulating feature activations, we follow the "all-at-once" method introduced in Bills et al. [1]. Unlike their original "one-at-a-time" approach—which requires one forwards pass per simulated token—the "all-at-once" method provides all the tokens to the model during one inference pass with the activations marked as `unknown`. The topk logprobs for each `unknown` can be used to compute the expected value of the associated token's activation. A more comprehensive discussion of this approach can be found in Bills et al. [1].

The original simulation method in Bills et al. [1] used the closed-source `text-davincii-003` model. Unfortunately, this model is deprecated, as is the associated `topk` logit feature for input tokens. To our knowledge, no cloud inference provider supports to `topk` logit feature, meaning that our simulation had to be carried out on local hardware.

We optimize local token simulation using a two-level key-value cache. Note that a simulation prompt can be broadly decomposed as

system prompt & few-shot examples + explanation + simulated sentence

The first level of the cache captures the system prompt and few-shot examples, which are shared across all explanations and simulated sentences. The second level appends the key-value store for the explanation. We can then use this to efficiently simulate the activations for a variety of different sentences for validation and testing.

We simulate all our explanations using the `gemma-2-27b-it` model loaded with four-bit quantization [34]. We selected this model as it significantly outperformed all equally or smaller sized models during our preliminary testing. Other models we considered included those from the Llama family [37], Mistral and Ministral families [14], and the distilled DeepSeek family [10]. The small size and compression of `gemma-2-27b-it` allows for local simulation on a single 40 GB Nvidia A100 GPU.

**Prompts.** The simulator prompts consist of three components. First, we provide the simulator with a header message, as shown in Figure 12. Next, we present the same set of few-shot examples that were shown to the explainer, formatted in the `token<tab>activation` format (Figure 8a). Finally, we supply the model with an explanation and a sentence, prompting it to predict the activation values for each token in the sentence using the message shown in Figure 13, with the sentence being simulated formatted as in Figure 8b. The predicted activation values are then collected using the "all-at-once" method described earlier.

```
We're studying neurons in a neural network. Each neuron looks for some
↪   particular thing in a short document. Look at summary of what the
↪   neuron does, and try to predict how it will fire on each token.

The activation format is token<tab>activation, activations go from 0 to 5,
↪   "unknown" indicates an unknown activation. Most activations will be 0.
```

Figure 12: The header message for the simulator, provided as a system message.

```
The previous messages were just examples. Now you will be given a sequence
↪   of tokens and asked to predict the activations of each token.

Always output a numerical activation, even if preceding activations are
↪   unknown. Even if you have predicted "unknown" many times previously,
↪   these were mistakes. Do not repeat this mistake, and output a
↪   numerical value from 0 to 5.
```

Figure 13: The prompt given to the simulator after the few-shot examples. Here we use the "all-at-once" method in Bills et al. [1] to collect all predicted activations values in a single inference pass.

### A.3 Complexity analysis

Our complexity analysis LLM judge is also locally executed and shares is architecturally similar to our simulation strategy. Instead of a bilevel key-value cache, we simply cache the system prompt and few-shot examples for the complexity judge once. We then reuse this cache to analyze every explanation *component* individually, before aggregating the total complexity as the mean over all components. As in the simulator, we prompt the complexity judge to provide an integer score between $0$ and $5$ for each component. Our model is `gemma-2-27b-it` with four-bit precision.

**Prompts.** The prompts for the complexity analyzer consist of three components. First, we provide a header message as a system message, as shown in Figure 14. Next, we present a set of few-shot examples demonstrating how to analyze the complexity of explanations. Each example includes an explanation component, an assigned integer complexity score, and a brief justification for the score. Finally, we prompt the complexity analyzer with a new explanation component and ask it to output an integer complexity score.

```
We're studying features in a neural network. Each feature has an
↪  activation rule describing how it activates.

Given an explanation of a feature's activation rule, predict how complex
↪  it is.

The complexity is a number between 0 and 5, where 0 is the simplest and 5
↪  is the most complex.

The complexity is determined by the level of abstraction required to
↪  understand the activation rule.

Having a specific activation context is more complex than not having an
↪  "Any" context.

Activation rules which activate only on a list of specific tokens are
↪  simpler than rules which activate on tokens capturing abstract
↪  concepts.
```

Figure 14: The system prompt for the complexity analyzer model.

The few-shot examples used for complexity analysis are as follows.

### Example 1

*Explanation component*: "the word 'instruments', specifically in a musical description, catalog, or reference"

*Complexity*: 1

*Justification*: `Low complexity - only activates on specific words.`

### Example 2

*Explanation component*: "present tense verbs ending in 'ing'"

*Complexity*: 2

*Justification*: `Moderate complexity - need to understand verb tenses and sp-ecific suffix patterns.`

### Example 3

*Explanation component*: "words related to medical conditions, in the context of movies and filmmaking"

*Complexity*: 3

*Justification*:  Higher complexity - due to need to recognize medical termi-
nology in metaphorical usage.

### Example 4

*Explanation component*: "The word 'risk', in the context of medical research/studies"

*Complexity*: 1

*Justification*:   Moderate complexity since only activates on specific words
in medical context.

### Example 5

*Explanation component*: "expressions of skepticism"

*Complexity*: 5

*Justification*: Very high complexity due to abstract nature of skepticism.

### Example 6

*Explanation component*: "words that reflect negative judgments",

*Complexity*: 5

*Justification*:  Very high complexity due to the abstract nature of negative
judgments.

# B    Holistic feature examples

In this section, we compare the regular and holistic activations for example sentences from our dataset, highlighting how holistic activations provide a clearer and more accurate characterization of a feature. As described in Section 4.1, holistic activations measure the contribution of each token to the feature's activation over the entire sequence. In contrast, regular activations reflect the feature's value at individual tokens, which can be difficult to interpret—especially for features that tend to activate only after a full concept has been completed, due to the causal structure of Transformer-based language models.

## B.1    The table cell separator feature

As an example, we examine the regular and holistic activations for the 1st feature at the 11th layer of the Gemma 2 9B model, which activates on table cell separators.

- *Regular activation*: 0 370 -33.764969 8.111195 93.7938
- *Holistic activation*: 0 370 -33.764969 8.111195 93.7938

Based on the regular activations alone, it appears that this feature activates on numbers. However, a closer look reveals that it actually fires on numbers that follow a table cell separator. The holistic activations successfully capture this pattern, offering a much clearer and more accurate interpretation of the feature's behavior.

## B.2    The mathematical operations feature

We present the regular and holistic activations on an example sentence corresponding to the 1st feature at the 21st layer of the Gemma 2 9B model, which activates on mathematical operations.

- *Regular activation*: int*); void increment_array (); int main (){
  increment_array(); } increment_address (int* ptr){  (*
- *Holistic activation*: int*); void increment_array (); int main (){
  increment_array(); } increment_address (int* ptr){  (*

From the regular activations on this sentence alone, it initially appears that this feature responds to pointers. However, the holistic activations reveal that it is specifically the operation of incrementing the pointer—not the pointer itself—that drives the feature's activation. This illustrates how holistic activations can more accurately capture the true behavior of a feature by considering its contribution to the overall sequence activation.

## B.3    The "do something like" feature

We show the regular and holistic activations on an example sentence for the 8th feature at the 31st layer of the Gemma 2 9B model, which activates on the phrase "do something like".

- *Regular activation*: to be able to do something like this: .xaml.cs: public partial
  class MyControl : UserControl {    public MyControl()
- *Holistic activation*: to be able to do something like this: .xaml.cs: public partial
  class MyControl : UserControl {    public MyControl()

The regular activations highlight the word "like", which alone does not clearly reflect the feature's behavior. However, examining the holistic activations reveals that this feature actually responds to the entire phrase "do something like", providing a much clearer interpretation of its activation pattern.

# C Complementary sentence examples

In this section, we present example features along with their structured explanations, top-activating samples, random non-activating samples, and top semantically similar non-activating samples. Activation values are visually highlighted using color, with darker shades indicating stronger activation.

For top-activating samples, we highlight their ground-truth activation values to illustrate the words and contexts that most strongly activate the feature. In contrast, for random and semantically similar non-activating samples, we highlight their simulated feature activations to assess the accuracy and potential errors of the generated explanations. Any positive simulated activation on these non-activating samples are *false positives*. We omitted the line breaks for readability.

We expect that overly broad explanations—those biased toward recall but lacking precision [4]—will produce more false positives on semantically similar samples, thereby revealing their inaccuracy.

## C.1 The "render" and "argument" feature

This is the 2nd feature on the 1st layer of the Gemma 2 9B model.

**Structured explanation**

```
[
    {
        "activates_on": "Words related to rendering in programming
        ↪  contexts, like 'render'.",
        "strength": 5
    },
    {
        "activates_on": "Words related to arguments and function calls
        ↪  within code.",
        "strength": 4
    },
    {
        "activates_on": "The word \"throws\" in a code context related
        ↪  to exception handling.",
        "strength": 4
    }
]
```

**Top-activating samples**

- render partial: 'edit_icon' %><h4><%= t('avatar.profile_change') %></h4>
- I have found that styled components, when not included in the initial render
- render h2 headline1`] = `.c0 { -webkit-letter-spacing:0.5px;-
- renderSubGame is called with an argument this.subGameInfo like so: renderSub
- .charts=render @time_series.chart, time_series: @time_series .charts=render
- title, @user.full_name.outer .container=render "/header", title: @user.full_name
- THead/index';import TR from '../TR/index';import { mount, shallow, render }
- namespace render { struct RenderState; };namespace scene { class PlaneNode :
- var Song = db.model('Song');var Album = db.model('Album');I want to render
- argument in call to a functionImagine a scenario like so: var obj = { render

**Random non-activating samples**

- joke, or is it serious?I'm watching two adaptations of Romeo and Juliet that take
- present invention relates to intumescent compositions based on foamed or
- 005-2013 Team XBMC  * http://xbmc.org  *  * This Program is free software;
- cases]. To investigate the application and efficacy of video-assisted thoracoscopic
- 2, y: 1, w: 1}. Give prob of picking 1 y, 1 v, 1 g, and 1 w
- 3125 How many millilitres are there in 37/5 of a litre? 7400 What is 1
- License, Version 2.0 (the "License");  * you may not use this file except in
- typemoq";import { IDirectoryManager, ISettingsManager } from "managers";
- 34445556667778889999"[i]; while ((i = getchar())
- def with_baton  until (baton = Baton.obtain) sleep(2)  end  result = yield

## Semantically similar non-activating samples

- "/components.html"/> <%def name="title()">${_("Global badges")}</%def>
- from "../js/src/components/component-list"; import { Metrics } from "../js/src
- a javascript method on a htmlwidget (jsoneditor) in shiny? I'm trying to use
- aser-example', width: 800, height: 600, scene: { create: create
- { useHistory } from 'react-router-dom'; import { Card, PageSection } from '@
- src="example_output.js"></script> <script> init = function() { var person =
- dom, args) { this.progress = 0.0; this.message = ""; this.dom = dom;
- call: puppeteer.launch().then(browser => { let html = ` <!DOCTYPE html>
- => { const { color, size, ...otherProps } = props  return React.createElement('
- == "undefined"){ _yuitest_coverage = {}; _yuitest_coverline = function(src, lin

As shown in the simulated feature activations above, the average number of false positives per sentence is significantly higher for semantically similar non-activating samples (11.6) compared to random non-activating samples (2.2). This discrepancy arises because the generated explanation describing "arguments and function calls within code" is overly broad. The Sentence Transformer effectively captures this, selecting coding-related samples that lead the simulator to generate more false positives.

## C.2   The table cell separator feature

This is the 1st feature on the 11th layer of the Gemma 2 9B model.

### Structured explanation

```
[
    {
        "activates_on": "Numerical values within specific contexts
        ↪  (e.g., scientific research, financial data, or technical
        ↪  documents).",
        "strength": 3
    },
    {
        "activates_on": "Specific HTML/data table markup",
        "strength": 2
```

```
    },
    {
        "activates_on": "The word \"Iris-setosa\"",
        "strength": 3
    }
]
```

**Top-activating samples**

- =\left(\begin{array}{rrrr|rr} -3 &3 &2 &2 & 0 & 0\\
-  0 & & & \\ & a & & \\ & & b & \\ &
- \ldots & x \\  x & x & a_3+x & \ldots & x \\ \vdots & \vdots& &\dd
- 0.999 0.999 5.1 Iris-setosa Iris-setosa 0.994
-  <path d="M14.59 8L12 10.59 9.41 8 8
- COUNT QTY UNIT OF MEASURE  Belgium Natural Gas Physical 5  412,500
-  Iris-setosa 0.996 0.996 4.6 Iris-setosa Iris-setosa
- 5.4 Iris-setosa Iris-setosa 0.998 0.998 4.4 Iris-
- 0.994 4.9 Iris-setosa Iris-setosa 0.991 0.991
- Positions 40     40  P&L Daily ($thousands) 0.5     (2.2)

**Random non-activating samples**

- * @name Includes per file  * @description The number of files directly included
- , appearances do matter. This is not a self-interested plea aimed at urging my
- 3 By Erin Roach NASHVILLE (BP) – The president of the Association of
- a) be the third derivative of -11*a**4/24 - 14*a**2. Let n(s) =
- int_{0}^{\sqrt{4-x^2-y^2}} z^2\sqrt{x^2+y^2+z^
- 10 // Search a list of files for lines that match a given regular-expression //
- Argonaute: A scaffold for the function of short regulatory RNAs. Argonaute is the
- import (  "os"  "syscall"  "unsafe" ) // Flags to control the terminals mode. const (
- to move freely from one point of connection to another in various networks it visits
- smoking program. This article has presented an overview of the Quit-Smoking

**Semantically similar non-activating samples**

- 00000;1.000000;1.000000;;  3.2000
-     - 0.34691048
-   - - 0.4292854
- 0.000837 0x001B // 0.000460 0x000
- 0.03744*-3 0.11232 0.003 * 3 0.009
- 1 2    1     2010-12-04 3    2     2010-12-01
- 0.07 times -1277 -89.39 0.85 * -0.03 -0.
- 0.2e" % 1.236  ' 1.24e+00' >» The total characters in the

- > <o1> <p1></p1> <p2></p2> <p3></p3
- 0 1943 Width: 1005 VWidth: 0 Flags: HM LayerCount: 2 Fore

As evident from the top-activating samples, this feature primarily activates on table cell separators. However, the explainer fails to capture this specificity, instead providing overly broad explanations such as "numerical values" and "HTML/data table markups." The Sentence Transformer is able to select samples related to numbers and HTML codes, leading the simulator to produce false positives. The average number of false positives per sentence for semantically similar non-activating samples is 18.7—significantly higher than the 2.9 observed for random non-activating samples.

## C.3 The mathematical operations feature

This is the 1st feature on the 21st layer of the Gemma 2 9B model.

**Structured explanation**

```
[
    {
        "activates_on": "Mathematical formulas, equations, and
        ↪   operations (e.g., sum, multiplication, division,
        ↪   exponential functions).",
        "strength": 5
    },
    {
        "activates_on": "Words related to lists (e.g., elements, length,
        ↪   sum)",
        "strength": 4
    },
    {
        "activates_on": "Mathematical functions (e.g., cos, sin, log,
        ↪   exp, arithmetic operations)",
        "strength": 3
    },
    {
        "activates_on": "Programming language constructs related to
        ↪   calculations (e.g., for loops, variables)",
        "strength": 2
    }
]
```

**Top-activating samples**

- mean of a list, i.e. the sum of all elements in the list divided by its length. (You
- code i get output as 392 #include<stdio.h> #define CUBE(r) ((r)*(r)*(r
- df}{dx} = \lim_{\Delta x \to 0} \dfrac{f(x + \Delta x) - f(x)}{\Delta
- true # let satisfies_associative_law = 1 + (2 + 3) = (1 + 2) + 3;;
- M[B](c => in (a => k(a) in c))    def map[B](f: A => B): M[
- the moment generating function of a random variable $X$ as: $E[e^{tX}]$? I know
- P(A \cap B) = P(B)\cdot P(A| B)=P(A)\cdot P(B|A)$, where P
- interest has compounded for a certain number of years.  Note: A = P(1+r)^t

- f is continuous at a number a if limit of f(x) as x approaches a is equal to f(a
- <stdlib.h> int max(int num1, int num2) {    int result;    if(num1 > num2

## Random non-activating samples

- Free Disk Space 46GB minimum Write Review Nioh is a Japanese game designed
- of code execution with wait and notify I research using wait and notify methods
- Omaha network will be down from 8:00 pm to 12:00 midnight, Tuesday,
- multiple racial origin and (2) the race categories of (2a) American Indian and
- * Copyright (C) 2009 Conexant Systems Inc.  * Authors <shu.lin@conexant.com
- All Rights Reserved.  * * Licensed under the Apache License, Version 2.0 (the
- 464230, -668516, -909964, -1188580?
- as possible. Mouse Control Click on a puzzle square to uncover the image or
- education. Since 1975 the flow of foreign medical graduates (FMGs) into US
- 0 Cal.App.2d 615 (1960) IRA GARSON REALTY COMPANY (a Corporation),

## Semantically similar non-activating samples

- I saw examples of my problem, but it seems I can't find correct loop. In any case,
- p1.x = x;   p1.y = y; } That's my code and the error I get is
- l := 0 * [1..6];  l[[1..3]] := 1; end; f(); Where();
- to the interval [0; 1]. The mapping should be deterministic in order to get the
- would be $1 = foo $2 = bar1 $3 = bar2 $4 = bar3 and so on.. it would be like re
- \{a<x<b\}$ can be written $[f(x)]^b_a$. Is there an equivalent notation for $f(x)
- WhereWithVars(); quit;  f:=function() if true = 1/0 then return 1; fi; return 2;
- #include <vector> #include <algorithm> using namespace std; int main()
- 2)^x, y=e^x) into a same coordinates.  I typed:  a = function(x){y=3^
- cin>>n; for(i=1;i<=10;++i) t=n*i; cout<<t<<endl;

From the top-activating samples, we observe that this feature primarily activates on mathematical operations such as summation, multiplication, division, and even limits. However, the generated explanation is overly broad, encompassing concepts like "mathematical formulas," "lists," "functions," and "programming language constructs." As a result, the Sentence Transformer, trained on top-activating samples, effectively selects math- and coding-related sentences, leading to a higher false positive rate. The semantically similar non-activating samples produce an average of 5.8 false positives per sentence, compared to just 0.9 for random non-activating samples.

## C.4   The definition feature

This is the 2nd feature on the 31st layer of the Gemma 2 9B model.

## Structured explanation

```
[
        {
```

```
                    "activates_on": "The word 'define' and related concepts
                    ↪    like 'defining' and 'defined'.",
                    "strength": 5
        },
        {

                    "activates_on": "The concept of explicit function or
                    ↪    struct declaration.",
                    "strength": 4
        },
        {

                    "activates_on": "The word 'Undefined'.",
                    "strength": 4
        }
]
```

**Top-activating samples**

- intl'; const messages = define Messages({ securityControlsLabel: { defaultMessage:
- Should I manually create a definition for GetEnumerator? Seems like it should
- DAU/MAU using Application Insights Analytics? Assuming I have a definition of
- Define Bessel's inequality $\$\$ \sum_{n=1}^{N}|a\_n|^2\leq \|x\|^2$
- : Undefined index in PHP post with AJAX In my local machine, I am trying to
- 4) & "'Sheet1'!A2" This gives me Error: 1004, Object-defined error. I'd li
- platform contributions (defined via org.eclipse.ui.popupMenus) in this perspective.
- want to define a metric with some labels but I don't always have them all the
- ) is defined as the condition of having abnormally small teeth \[[@B1]\]. According
- but definition of anacronym is sketchy : http://bit.ly/hJtAq4 Richard@Home

**Random non-activating samples**

- I've had the idea of the WWP lettering shadows for a while in my head, but just
- men under the Diocese of Australia and New Zealand, ROCOR. The monastery is
- are being studied as microdosemeters since they can provide sensitive volumes of
- idermal surface saccharides reactive with phytohemagglutinins and pemphigus
- Deviates from the original intent"? I suggested this edit yesterday: the question is
- and Adaptive Reactions in People Treated for Chronic Obstructive Pulmonary
- Feb. 23, 2014 (HealthDay News) – Preteens who changed schools frequently when
- this file except in compliance with the License. # You may obtain a copy of the
- *************************************************/ /* ATutor */ /**
- link a jQuery UI sortable element with an array? So, basically, how can I link a

**Semantically similar non-activating samples**

- > <o1> <p1></p1> <p2></p2> <p3></p3
- _list.hpp> boost::adjacency_list<boost::vecS, boost::vecS, boost::directedS,

- of 1 micron. Its interior mechanism rests on a heavy cast iron base that is covered
- class AB amplifier: - Rv adjusts the bias point of the two transistors so
- /mkcharacters share/texmf-dist/scripts/luaotfload/mkglyphlist share/texmf-dist
- if (is_android) { } declare_args() { - compile_suid_client = is_linux +
- Reference #9.44952317.1507271
- Application No. 2000-159163, filed Mar.
- NppExecute plugin in notepad++. I am not able to figure out next step to
- a customer can see, based on their customer number and order type. I was using

As shown in the top-activating sentences, this feature primarily activates on the words "define" and "definition". However, the generated explanation broadens this to include "concepts of explicit function or struct declarations", which proves overly inclusive. The similarity-based complementary sentences contain code snippets involving declarations, leading the feature simulator to produce false positives. On average, the semantically similar non-activating samples trigger 1.6 false positives per sentence, significantly higher than the 0.3 observed for random non-activating samples.

# D  Supplemental experimental results

## D.1  Supplemental explainer comparison results

The tabular results of Figure 4 are shown in Table 1.

Table 1: The explainer comparison results in tabular form.

| Model | One-shot Unstructured | One-shot Structured | One-shot Structured w/ Negatives | One-shot Structured w/ Holistic | Tree Unstructured | Tree Structured | Tree Structured w/ Negatives | Tree Structured w/ Holistic |
|---|---|---|---|---|---|---|---|---|
| Gemma 2 9b | $0.363 \pm 0.032$ | $0.384 \pm 0.032$ | $0.366 \pm 0.033$ | $0.370 \pm 0.032$ | $\mathbf{0.435} \pm 0.030$ | $0.431 \pm 0.032$ | $0.430 \pm 0.032$ | $0.432 \pm 0.032$ |
| Llama 3.1 8b | $0.332 \pm 0.033$ | $0.360 \pm 0.033$ | $0.342 \pm 0.034$ | $0.349 \pm 0.034$ | $\mathbf{0.404} \pm 0.032$ | $0.392 \pm 0.034$ | $0.389 \pm 0.034$ | $0.399 \pm 0.033$ |
| GPT-2 Small | $0.520 \pm 0.033$ | $0.568 \pm 0.031$ | $0.557 \pm 0.032$ | $0.526 \pm 0.033$ | $0.622 \pm 0.028$ | $0.626 \pm 0.028$ | $0.623 \pm 0.028$ | $\mathbf{0.634} \pm 0.028$ |

## D.2  Complementary sentence sourcing correlation scores

We present the correlation scores for the four complementary sentence sourcing methods in Figure 15. As shown in Figure 15, both the one-shot and tree explainer yield consistently lower correlation scores when evaluated on similarity-based complementary sentences. This further highlights the recall bias of existing feature explanation methods and demonstrates the effectiveness of similarity-based strategies in identifying "close counterexamples."

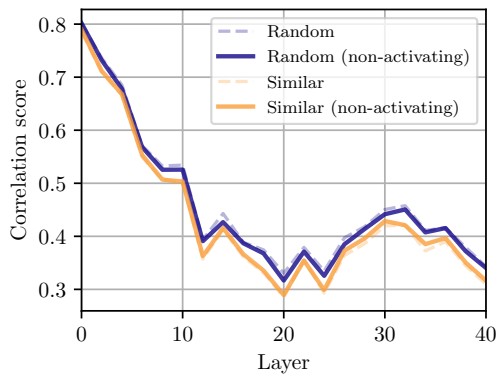
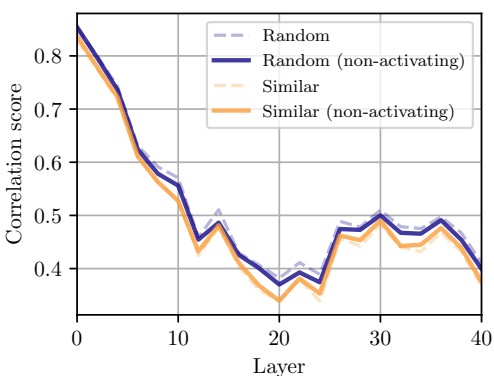

(a) Results for the one-shot explainer.    (b) Results for the tree explainer.

Figure 15: The correlation score between the simulated and ground-truth activations, averaged over 50 features per layer. "Non-activating" indicates that the sentences have no ground-truth feature activation.

## D.3  Feature complexity and polysemanticity for the one-shot explainer

Figure 16 presents the complexity and polysemanticity plots for explanations generated by the one-shot explainer. Across all three models, we observe a general upward trend in complexity as depth increases. Meanwhile, polysemanticity tends to peak in the middle layers, remaining lower in both the early and final layers.

## D.4  Supplemental results for the structured explanations

In Figure 17, we plot the correlation scores of structured explanations generated by the one-shot explainer across different layers and numbers of rules (i.e., explanation components). For each rule count $i$ from 1 to 5, we report the proportion of the maximum score achieved by explanations with up to $i$ rules, relative to the overall maximum score across all rule counts. As shown, the correlation score rises rapidly as more rules are included in the structured explanation, highlighting the benefit of combining multiple monosemantic rules. This trend helps explain the superior performance of structured over unstructured explanations for the one-shot explainer in Figure 4.

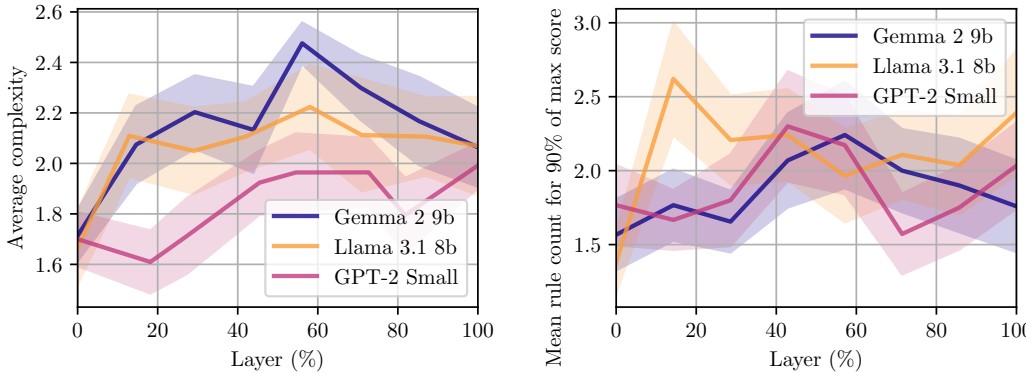

Figure 16: Lines denote mean values, and shaded regions denote 80% confidence intervals. The horizontal axis corresponds to the analyzed layer as a proportion of the total layer count, which varies between models. **Left.** The average explanation complexity for the one-shot explainer. **Right.** The polysemanticity of structured explanations generated by the one-shot explainer.

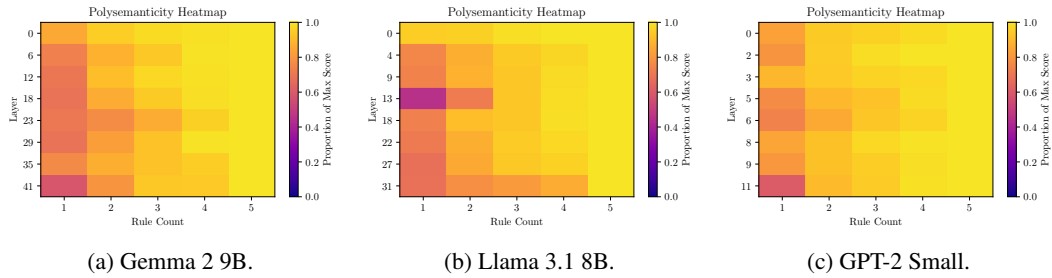

(a) Gemma 2 9B.  (b) Llama 3.1 8B.  (c) GPT-2 Small.

Figure 17: The correlation scores of structured explanations generated by the one-shot explainer across different layers and number of rules.

We present the same analysis for the tree-based explainer in Figure 18. Here, the performance differences between various maximum rule counts are less pronounced. This is consistent with Figure 4, where structured explanations offer little advantage over unstructured ones when using the tree-based explainer. We hypothesize this is because the tree-based explainer's iterative refinement process naturally incorporates the feature's polysemantic behaviors into its explanations, reducing the need for explicit structuring.

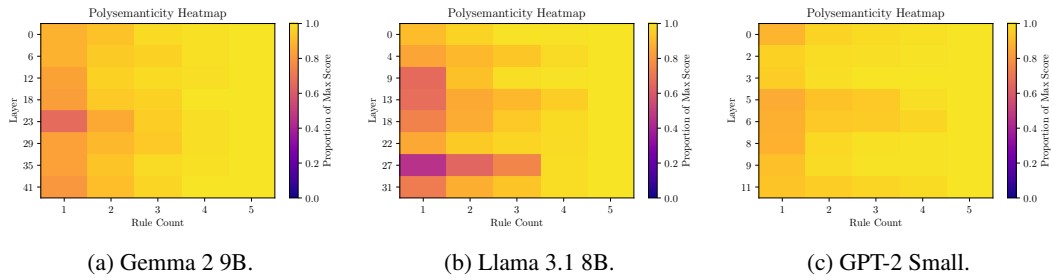

(a) Gemma 2 9B.  (b) Llama 3.1 8B.  (c) GPT-2 Small.

Figure 18: The correlation scores of structured explanations generated by the tree-based explainer across different layers and number of rules.

# E   Licenses

We use a subset of the Pile which eliminates potentially copyrighted content [8, 25]. The Pile itself is MIT-licensed. GPT-2 is released under the Apache license. Llama 3.1 and Gemma 2 licenses are permissive [23, 20].

