# OpenReview forum: "Revising and Falsifying Sparse Autoencoder Feature Explanations"
_NeurIPS.cc/2025/Conference — NeurIPS 2025 poster_

### Official Review · Reviewer_LBic · 2025-06-26

**Clarity:** 3
**Significance:** 3
**Originality:** 3
**Rating:** 4
**Confidence:** 4

**Summary:**

This paper attempts to address overly broad explanations produced by autointerpretability methods by introducing a strategy to falsify generated explanations through sourcing similar sentences as negative samples, hence making them more precise. In addition, the authors propose structured scheme to produce fine-grained explanations along with tree-based method for explanation refinement. The experiments confirm the bias of current autointerpretability methods toward recall at the cost of precision, and also show how the structured explanation scheme and tree-based refinement can improve generated explanations. Lastly, the authors provide analyses of how complexity and polysemanticity of the produced explanations evolve across layers by using an external LLM (instruction-tuned Gemma2 27B) as a judge.

**Questions:**

- L44: Explanatations → Explanations?
- L25: Missing citation to [(Bolukbasi et al., 2021)](https://arxiv.org/abs/2104.07143).
- L200: You mentioned that the explainer LLM is prompted such that each component corresponds to a monosemantic concept. Is there any guarantee for that?
- Do you have any conjecture regarding how different width of SAEs might affect the quality of explanation for polysemantic feature?

**Ethical Concerns:**

["NO or VERY MINOR ethics concerns only"]

**Limitations:**

Please refer to the weakness section.

**Paper Formatting Concerns:**

font style and size seem to deviate from official template.

**Quality:**

3

**Strengths And Weaknesses:**

**Strengths:**
- The issue of how autointerpretability methods are heavily biased towards recall is relatively understudied. This paper addresses a crucial gap in current approaches by proposing similarity-based method for sentence sourcing to evaluate feature explanations.
- Using a tree-based jailbreaking technique to automatically generate and refine explanations is a valuable contribution. Although it is computationally expensive as alluded in the limitation section, the improvement over one-shot generation is quite significant.
- The proposed methods in the paper are simple to implement, making it easily applicable in various settings.

**Weaknesses:**
- There is no human evaluation to confirm whether the proposed methods improve automatic interpretability of SAEs in a meaningful way. Furthermore, using LLM as a judge potentially introduce another biases that could affect evaluation reliability, particularly for concepts across different languages or domains.
- All experiments are conducted on an uncopyrighted subset of the Pile, which may not fully represent all possible concepts that a model might encode. A more detailed discussion on how the dataset choice might affect the results would be beneficial.
- While features can be polysemantic as defined in L46 and L201, more thorough explanations of how this can be different from the property of features in SAE latent space that is known to be more monosemantic and interpretable (L31) is needed, to provide readers with clarity.

---

> ### Author Rebuttal · Authors · 2025-07-31
>
> We thank Reviewer LBic for the constructive review. We address your concerns as follows.
>
> ---
>
> **W1.** There is no human evaluation to confirm whether the proposed methods improve automatic interpretability of SAEs in a meaningful way. Furthermore, using LLM as a judge potentially introduces biases that could affect evaluation reliability, particularly for concepts across different languages or domains.
>
> **A1.** Regarding human evaluation, evaluating all features would be prohibitively expensive. However, we have performed extensive case studies across various aspects. We provide examples of semantically similar negatives falsifying SAE explanations in Appendix C, tree-based explainer generating higher-quality explanations in **A1** of our response to Reviewer gHHG, and structured explanations capturing polysemantic concepts in **A4** of that same response. These case studies confirm our methods generate more accurate explanations for SAEs. We have also provided code that facilitates reproduction and manual inspection of our results.
>
> While LLM judges can introduce bias, we don't use them in our main experiments. Explanation performance in Section 5.2 is measured by the correlation coefficient between simulated and ground-truth activations, consistent with previous work [1]. We only use an LLM judge when measuring feature complexity, where we use carefully-written few-shot examples and confirm through manual inspection that the LLM's judgments align with human intuition.
>
> **References:**
>
> [1] Bills et al. Language models can explain neurons in language models. *OpenAI blog*.
>
> ---
>
> **W2.** All experiments are conducted on an uncopyrighted subset of the Pile, which may not fully represent all possible concepts that a model might encode. A more detailed discussion on how the dataset choice might affect the results would be beneficial.
>
> **A2.** Dataset choice can indeed affect results. However, the Pile is large, and our 100,000 samples should capture the vast majority of concepts in SAE features. If a concept encoded by an SAE feature were absent from the dataset, we would observe no activations—yet this never occurred in our evaluated features. All features in our experiments showed a significant number of activating sentences, making our results meaningful.
>
> ---
>
> **W3.** While features can be polysemantic as defined in L46 and L201, more thorough explanations of how this can be different from the property of features in SAE latent space that is known to be more monosemantic and interpretable (L31) is needed, to provide readers with clarity.
>
> **A3.** Thank you for this suggestion. In L31, SAE features are described as monosemantic and interpretable relative to neurons. The large latent dimension of SAEs allows us to decompose polysemantic neurons into more monosemantic concepts. However, the typical latent dimensions in our experiments (16k to 64k) are still far fewer than the concepts encoded in the LLMs' latent space. Therefore, SAE features must still be inherently polysemantic, making it essential to break down this polysemanticity for accurate explanations. By representing polysemantic concepts in a structured format, we obtain a clear measure of feature polysemanticity and improve explanation quality.
>
> ---
>
> **Q1.** Typos and missing citations.
>
> **A4.** Thanks for pointing these out. We'll fix the typos and add the missing citations.
>
> ---
>
> **Q2.** L200: You mentioned that the explainer LLM is prompted such that each component corresponds to a monosemantic concept. Is there any guarantee for that?
>
> **A5.** While there is no hard guarantee, we observe that during the tree-based explainer's iterative refinement, it tends to break down polysemantic concepts into monosemantic components. Explanations with clear, monosemantic components achieve higher validation scores and survive the pruning process. Through case studies (e.g., **A4** in our response to Reviewer gHHG), we find this pattern holds across all inspected features.
>
> ---
>
> **Q3.** Do you have any conjecture regarding how different width of SAEs might affect the quality of explanation for polysemantic feature?
>
> **A6.** This is an insightful question. SAEs are trained with a sparsity loss in the feature space. As SAE width increases, they tend to decompose polysemantic features into more monosemantic components, a phenomenon known as feature splitting [1]. Therefore, we believe SAEs with higher widths would exhibit lower polysemanticity. However, since the latent dimension of SAEs cannot exceed the number of concepts in natural language, polysemanticity will always exist in SAE features, necessitating better handling methods. Our proposed structured explanations better capture this polysemanticity, resulting in consistent performance improvements.
>
> **References:**
>
> [1] Chanin et al. A is for Absorption: Studying Feature Splitting and Absorption in Sparse Autoencoders. *arXiv preprint arXiv:2409.14507*.
>
> ---
>
> We hope our response addresses your concerns. If you have further questions, we're happy to address them in the discussion period.

---

> ### Author Response · Authors · 2025-08-06
> **Follow-up on Our Rebuttal**
>
> Dear Reviewer LBic,
>
> Thank you once again for your thoughtful and constructive feedback on our submission. We deeply appreciate your engagement during the review process and are grateful for your positive assessment of our work.
>
> As the discussion phase is drawing to a close, we wanted to kindly confirm whether there are any remaining concerns or clarifications we can provide. We are happy to respond to any further questions you may have.
>
> Thank you again for your time and support.
>
> Best regards,
>
> The Authors

---

> > ### Comment · Reviewer_LBic · 2025-08-06
> >
> > Thank you for the clarifications. I would suggest the authors to incorporate the raised points, especially A3. While I agree that human evaluation for all features can be prohibitively expensive, I believe that taking several proxy features should be possible and would significantly strengthen the contribution of this work. I maintain the current score as I believe it is already appropriate.

---

> > > ### Author Response · Authors · 2025-08-06
> > > **Response to Reviewer LBic**
> > >
> > > Thank you very much for your thoughtful feedback and for taking the time to read our paper and provide a constructive review. We will ensure that the raised points, especially A3, are thoroughly incorporated into the revised version of our paper.
> > >
> > > We agree that taking proxy features in place of full-scale human evaluation would further strengthen our contribution. In this direction, we have already conducted extensive sanity checks across multiple components of our method—including the semantically similar negatives, the tree-based explainer, the structured explanations, and the LLM complexity analyzer—to ensure alignment with human intuition. These examples, shared in our responses to Reviewers gHHG and i1y8, will be integrated into the final version of the paper.
> > >
> > > Once again, we sincerely thank you for your helpful suggestions and for maintaining a positive score, which we greatly appreciate.

---

### Official Review · Reviewer_i1y8 · 2025-06-26

**Clarity:** 3
**Significance:** 2
**Originality:** 2
**Rating:** 4
**Confidence:** 4

**Summary:**

The paper studies feature explanations of Sparse AutoEncoders (SAEs) by using LLMs. For this, the paper builds on the idea of Bills et al. (2023) to use an iterative approach to explain individual neurons of SAEs by prompting an LLM. By proposing a tree-based explanation method and a structured, component-based format, the paper tackles challenges of Bills et al. (2023) such as overly broad explanations. Additionally, the paper analyzes the evolution of feature complexity and polysemanticity across layers of different models. The claims and statements of the paper are backed-up by empirical results by prompting LLMs.

**Questions:**

1. In the abstract it is stated the tree-based method "refines explanations". This is assessed in 5.3 (Feature Complexity) by using an LLM judge. How can we sure that LLM judge does a good job? In the current form, the assessment of explanations happens by a metric (the LLM judge) that we don't understand.
2. What does "close counterexample" mean? How is closeness mathematically defined?
3. Section 4.2, Branching step: The creation of the child explanations is without a score. How is it possible then to perform the pruning in step 3? It would be beneficial to formalize the methods in Section 4 by the notation of Section 3.2 (pseudo code). This would help readability and would better connect the different sections of the paper.
4. Does no ground-truth mean zero activation? I found this confusing in the paper. Also the statement in L213 is equivalent to $f*(\mathbf{t})_{i}=0,\forall i$ under the restriction that all values have to be positive, which is more difficult to grasp from the presented equation.
5. L217: Random non-activating sentences: "defined as the proportion of positive elements". This means that the value is in [0,1]. However, looking at Fig.3 it is above 1. How is this possible?
6. L306: Is there any evidence for this hypothesis?

**Some comments unrelated to my assessment of the paper:**

- The wording "Feature Learning" and, at some place in the paper, "Learning" is misleading since there is no learning in an ML sense involved. It is prompting of an LLM.
- L112: $f(t)$ not $f(\mathcal{T})$; by definition f **maps** elements from $\mathcal{T}$
- L115: **an** explanation
- L120: What constraint?
- The wording of the paper should be improved. For example,
   - L183: I wouldn't call this causally as causal has a strong meaning and clear definition in mathematics that I doubt to hold here.
   - The use of training where inference or prompting is meant.
- L146: I guess it must be "negative log probabilities".
- L257: The font type of LLama 4 Scout is different than the others.

**Ethical Concerns:**

["NO or VERY MINOR ethics concerns only"]

**Final Justification:**

I appreciate the authors response and the entire rebuttal. I will increase my score to **borderline accept** assuming that the authors will apply the promised changed.

During the rebuttal, I have read the reviews of the other reviewers that evaluate the contribution (the improvement of Bills et al. approach) as significant and valuable. Hence, I have reconsidered my assessment and agree that this an interesting contribution for the community.

My change in the assessment is also based on the promises of the authors to change the paper as discussed during the rebuttal (e.g., presentation of pseudo code). Moreover, I urge the authors to realistically state their contributions and to not oversell it unrealistically (e.g., overselling of "understanding" or "case study"). A realistic reflection of the contributions and limitations is appreciated by the community and the conference to know in advance what to expect and where are points for improvement. Maybe the usage of the limitations section is a good place for that.

Once more, I thank the authors for the good rebuttal and wish all the best.

**Limitations:**

Yes, the limitations discussion is already good. However, it could be extended in the direction of how this method helps to understand LLMs. For instance, the second point mentions polysemanticity: What is the consequence if this fails?

**Paper Formatting Concerns:**

The font type is slightly different to other papers I have reviewed for NeurIPS this year.

**Quality:**

2

**Strengths And Weaknesses:**

### Strengths

- **Clarity:** The paper is well-written, well-organized and easy to follow.
- **Clarity:** The motivation of the paper is clear (see summary).
- **Clarity:** The mathematical formalization in 3.2 is good.
- **Originality:** The paper clearly describes how the proposed method is different to the original work of Bills et al. (2023).

### Weaknesses

- **Originality:** The paper seeks to advance the field of XAI, in particular, understanding LLMs. Method-wise, it is clear how the method advances over the original work of Bills et al. (2023). However, it is not clear how the proposed method helps in understanding LLMs compared to Bills. In the abstract it is stated that "this work addresses these limitations [overly broad explanations and polysemanticity]" but the results demonstrate that these limitations are not addressed (e.g., see statement L316-317) and only marginally quantitatively improved (see figures and tables)---completely ignoring the true goal of XAI on enhancing the understanding of LLMs.

- **Significance:** The contribution of the paper is marginal as it is mainly the approach of Bills et al. (2023) with a new prompting strategy.

- **Quality:** After reading the paper, I'm not sure about the take away of the paper accept from what was already know from Bills et al. (2023): you can explain an LLM with an LLM to some extent but it might fail for unknown reasons. Moreover, the approach the paper builds upon is the explanation of a black-box with a black-box (a notoriously ill-defined task; e.g., see [1]). Instead of presenting numerical results, it would be beneficial to show how this approaches enhances our knowledge about LLMs (similar to the presentation of Bills et al.). For instance, why does a model hallucinate or, simpler, how polysemanticity of neurons impacts text generation. I noticed that the appendix provides some illustrative examples but a thoroughly discussion and concise conclusion is missing.

- **Quality:** Certain choices of design decisions are not well-justified. For instance, why is the max-norm a good choice for the ordering, why a random sample of top quantile, or how is top defined. In order to understand the sensitivity of the proposed method it is required that the paper presents some kind of ablation study or discusses the selection of these hyperparameters. Otherwise it is not clear whether the observed results are only good by chance or cherry-picked. Another example is the usage of cosine similarity where results suggest that this could be inappropriate [2].

- Minor issue **Quality**: References should be to publications not arXiv (e.g., [4])

I will elaborate on a few more things in the questions section.

[1]: Cynthia Rudin: Stop Explaining Black Box Machine Learning Models for High Stakes Decisions and Use Interpretable Models Instead. 2018.

[2]: Steck et al.: Is Cosine-Similarity of Embeddings Really About Similarity? 2024.

---

> ### Author Rebuttal · Authors · 2025-07-31
>
> We thank Reviewer i1y8 for the constructive review. However, we believe there are some misunderstandings regarding our paper. We'll address your concerns as follows.
>
> ---
>
> **W1.** However, it is not clear how the proposed method helps in understanding LLMs compared to Bills. […]
>
> **A1.** Thank you for the comment. We respectfully disagree that our paper does not help with understanding LLMs. In fact, our work improves this understanding in several important ways:
>
> 1. **Improving explanation accuracy matters.** SAE research aims to understand LLMs by interpreting their features. But if those explanations are inaccurate—too broad or missing polysemanticity—they hinder, rather than help, our understanding. Many current auto-interpretability methods suffer from these issues [1,2]. Our work directly addresses them.
> 2. **We are the first to tackle overly broad explanations.** Prior methods (like [1]) use evaluation metrics that don’t penalize overly inclusive explanations. This allows inaccurate, misleading outputs. Our method introduces *semantically similar negative sentences* to detect and fix overly broad explanations. Figure 3 shows that our tree-based explainer reduces this problem through iterative refinement.
> 3. **We are the first to systematically capture polysemanticity.** Polysemanticity—when a feature activates on unrelated concepts—has been a known problem in XAI. We use a structured explanation format that not only captures this property but also lets us quantify it. In experiments, structured explanations consistently outperform unstructured ones across datasets and settings (see responses to Reviewer q24V and A4).
> 4. **We analyze how feature complexity and polysemanticity evolve.** Using our methods, we examine how these properties change across layers and models (Figure 5). For example, middle layers tend to encode more complex concepts, and stronger models (GPT-2 < Gemma 2 < Llama 3.1) have more complex and polysemantic features. These patterns deepen our understanding of LLM internals.
>
> Regarding the reviewer’s point about L316-317: those lines refer only to the limits of holistic activations and training negatives, which we report as negative results. The new methods we introduce—semantically similar negatives, structured explanations, and the tree-based explainer—directly address these problems and show clear performance gains (Figures 3 and 4).
>
> In short, our paper directly tackles major open issues in SAE interpretability and provides tools that improve both explanation quality and our understanding of LLMs.
>
> **References:**
>
> [1] Bills et al. Language models can explain neurons in language models. *OpenAI blog*.
>
> [2] Paulo et al. Automatically Interpreting Millions of Features in Large Language Models. In *ICML*.
>
> ---
>
> **W2.** The contribution of the paper is marginal as it is mainly the approach of Bills et al. with a new prompting strategy.
>
> **A2.** Our approach goes well beyond just a new prompting strategy. The tree-based explainer automatically improves explanations using feedback from validation sets. It produces structured, intuitive explanations and introduces a way to detect overly broad features using semantically similar negative sentences—something not done before. These methods are part of an iterative refinement process that significantly improves explanation accuracy and helps us better understand LLMs.
>
> Also, our contribution is not marginal compared to Bills et al. That work focuses on generating and evaluating SAE feature explanations but does not address major issues like overly broad features and polysemanticity. As a result, their explanations are often inaccurate. In contrast, we directly tackle these issues and show clear improvements (see our response to Reviewer gHHG). Finally, while Bills et al. is mainly methodological, our paper also analyzes how complexity and polysemanticity change across layers and models, offering deeper insight into LLM behavior (see Figure 5).
>
> ---
>
> **W3.** After reading the paper, I'm not sure about the take away of the paper except from what was already know from Bills et al. […]
>
> **A3.** We address how our paper goes beyond Bills et al. in **A1**, but to summarize: our work is the first to systematically tackle the issue of overly broad and polysemantic SAE features. Unlike Bills et al., which focuses mainly on methodology, we use our tools to study the internal structure of LLMs. For instance, we show that middle layers tend to encode more complex and polysemantic concepts, and that stronger models (e.g., GPT-2 < Gemma 2 < Llama 3.1) have more such features. This directly connects to the reviewer’s question—polysemanticity helps models represent more concepts per neuron, making them more capable.
>
> We also understand the concern about using one black box to explain another. In our main experiments, though, we don’t rely on LLM judges—explanations are evaluated using correlation with ground-truth activations, following Bills et al. We do use an LLM judge to estimate feature complexity, but we design high-quality few-shot prompts and find that its judgments align well with human intuition.
>
> ---
>
> **W4.** Certain choices of design decisions are not well-justified.
>
> **A4.** We justify our design choices as follows:
>
> - We use maximum activation to rank sentences so that a few repeated activating words don’t dominate the score.
> - We sample randomly from the top quantile instead of just taking the top-activating sentences to better capture polysemantic features. Looking only at the top examples can make a feature seem monosemantic. For example, the 1st feature in Gemma 2 9B seems to focus on "instrument," but looking deeper reveals it also activates on "warn" and "warning." Sampling more broadly helps uncover this.
> - As for how we define "top," we split the dataset into 1000 activation quantiles and sample from the top one. To address concerns about cherry-picking, we ran experiments on different quantiles on the 10th layer of Gemma. In all cases, the tree-based explainer consistently outperforms the one-shot one.
>
> | #quantiles | One-shot score | Tree score |
> | --- | --- | --- |
> | 100 | 0.230 | 0.282 |
> | 500 | 0.356 | 0.376 |
> | 1000 | 0.428 | 0.482 |
> | 2000 | 0.387 | 0.493 |
> | 3000 | 0.456 | 0.513 |
>
> ---
>
> **Q1.** In the abstract it is stated the tree-based method "refines explanations". This is assessed in 5.3 (Feature Complexity) by using an LLM judge. How can we sure that LLM judge does a good job?
>
> **A5.** Thank you for the question. We believe there may be a misunderstanding. Section 5.3 is not used to evaluate the quality of explanations from the tree-based explainer. That evaluation is done in Section 5.2, where we use the correlation between simulated and actual activations—just like in Bills et al.
>
> Section 5.3 instead studies how feature complexity changes across layers and models. Here, we use an LLM judge, but we design high-quality, human-written few-shot prompts to guide it. Based on manual checks, the LLM’s ratings align well with human intuition.
>
> ---
>
> **Q2.** What does "close counterexample" mean? How is closeness mathematically defined?
>
> **A6.** "Close counterexamples" are sentences that are semantically similar to the top-activating sentences but on which the SAE feature doesn't activate. They are useful for exposing overly broad explanations, as shown in Figure 3. The definition involves semantic similarity which is challenging to describe mathematically.
>
> ---
>
> **Q3.** The creation of the child explanations is without a score. How is it possible then to perform the pruning in step 3?
>
> **A7.** The child explanations are scored on a validation set, and these validation scores are used for pruning. This process is iterative: at each iteration, we generate variations of current explanations, score them on validation sets, then prune the explanations using validation performance. We'll formalize this into pseudo code in our paper to improve clarity.
>
> ---
>
> **Q4.** Does no ground-truth mean zero activation?
>
> **A8.** Yes, no ground-truth means the SAE feature has zero activation on the sentence.
>
> ---
>
> **Q5.** […] However, looking at Fig. 3 it is above 1. How is this possible?
>
> **A9.** The value in Figure 3 refers to the number of false positives in the sentence. To get the ratio of false positive elements, we need to divide it by the length of sentences.
>
> ---
>
> **Q6.** L306: Is there any evidence for this hypothesis?
>
> **A10.** On L306, we hypothesize that the tree-based explainer is capable of packing polysemantic meanings into a single explanation. This is mainly derived from case studies. We provide examples of one-shot and tree-based explanations below, where the one-shot explanation misses polysemantic components and achieves a lower score, while the tree-based explainer successfully packs polysemantic concepts into a single string.
>
> **Example 1:**
>
> - One-shot (score: 0.738): instruments and related equipment or devices
> - Tree (score: 0.894): The neuron is looking for mentions of specific types of devices, tools, or equipment, particularly when referred to by their common names or technical terms, such as "instruments", "instrument", or warning messages like "warnings" or "Warning".
>
> **Example 2:**
>
> - One-shot (score: 0.126): references to software licenses and technical documentation metadata, such as version numbers and programming terms.
> - Tree (score: 0.270): The neuron is looking for specific keywords related to licensing, particularly "License" and "Apache License", as well as some LaTeX-related keywords like "article" and specific package names. It also activates on certain version numbers and technical terms like "time" in the context of Java imports.
>
> ---
>
> We hope our response addresses the reviewer's concerns and would appreciate your consideration in raising the score. Should you have any further questions, we welcome the opportunity to address them during the discussion period.

---

> > ### Comment · Reviewer_i1y8 · 2025-08-01
> > **Comment on A1**
> >
> > Thank you for providing such a detailed rebuttal. I really appreciate that and I have to think about a few things.
> >
> > Regarding A1: I have not questioned that the method improves over the SOTA (Bills et al.) method-wise. I have questioned that it helps in understanding LLMs compared to SOTA. This is an important difference. The points mentioned in A1 don't address how this helps in understanding LLMs. We apply an XAI technique to a method to reveal information and, in the best case, information provides knowledge, hence, helping us to understand the method. Using this knowledge (or the understanding), we can answer questions about a methods behavior such as: "Why did the model fail on this particular case?", "Why does a method hallucinate?", or "What can I do to improve my method?".
> >
> > So again my question: How does the method help in understanding LLMs compared to Bills? This is the point I stressed in W1. This also relates to W3: "After reading the paper, I'm not sure about the take away of the paper accept from what was already know from Bills et al. (2023): **you can explain an LLM with an LLM to some extent but it might fail for unknown reasons.**"
> >
> > Looking forward to hearing your thoughts.

---

> > > ### Author Response · Authors · 2025-08-02
> > > **Response to A1 and A2 (1/3)**
> > >
> > > We thank Reviewer i1y8 for the prompt response and insightful comments. The reviewer raises an important point, not only for our paper but for the field of XAI more broadly. In response, we aim to clarify that XAI research on SAEs goes beyond generating human-appealing explanations—it also contributes to a deeper, mechanistic understanding of how LLMs process information. We will present relevant literature demonstrating how SAE-based explanations have been used to uncover internal structures and behaviors within LLMs, and explain why XAI research in this domain avoids some of the key pitfalls observed in computer vision. Finally, we will include examples focused on LLM refusal behavior, highlighting cases where the method of Bills et al. can produce misleading explanations, while our approach avoids such issues.
> > >
> > > > The race to generate human-appealing (more accurate) explanations led us (more or less) nowhere. Hence, I urge the authors to demonstrate that the proposed method indeed increases our understanding of LLMs, for instance, by **demonstrating that we can do something useful with it beyond the generation of human-appealing explanations** and, hence, avoiding the pitfall from computer vision.
> > >
> > > We agree with the reviewer that the fundamental goal of XAI research is to understand how LLMs work, rather than simply producing human-appealing explanations. To underscore that our work has implications beyond visualization—and that SAE-based XAI is heading in a productive direction—we highlight several recent studies where SAE explanations have been essential for understanding and influencing LLM behavior:
> > >
> > > **1. Reducing hallucinations using SAE feature explanations**
> > >
> > > The reviewer suggested hallucination as a promising application. [1] is a relevant work that uses SAE explanations to both study and mitigate hallucinations. The SAFE system proposed in [1] consists of two stages. The first stage detects hallucination using entropy-based heuristics (not involving SAEs). In the second stage, the authors apply SAE enrichment by identifying and emphasizing **context-relevant SAE features**, while filtering out irrelevant or misleading ones. These feature-level **explanations** are used to enrich the prompt, making it crucial that the explanations are accurate. Across three QA datasets, SAFE improves accuracy by up to 29.5% and significantly reduces hallucination. This directly informs the reviewer’s question of **why LLMs hallucinate**: since hallucination decreases when enriching the prompt with relevant SAE-explained features, it suggests hallucinations stem from reliance on irrelevant internal representations—representations which can be diagnosed and identified using **SAE explanations** (see Table 3 of [1]).
> > >
> > > **2. Understanding the reasoning behavior of LLMs**
> > >
> > > [2] examines reasoning in LLMs post-trained with reinforcement learning. The authors identify “reasoning features” via an SAE trained on a chain-of-thought model, and find that amplifying these features improves reasoning accuracy. Although [2] identifies reasoning features using a “reasoning vocabulary,” we believe this process could be made more efficient and robust by leveraging the **feature explanations** from a pre-trained SAE—if those explanations can be generated more accurately than with current methods.
> > >
> > > **3. Steering refusal features to improve model safety**
> > >
> > > [3] uses an SAE to discover “refusal” features in a Phi-3 Mini model. By amplifying those features at inference time, they significantly increase the model’s refusal rate on unsafe/jailbreaking prompts. This demonstrates that SAE explanations could be used to steer the model to a safer state: a direct answer to “how we can improve model behavior.”
> > >
> > > In summary, these examples show that SAE-based XAI is not just about interpretability for its own sake. Rather, it enables practical insights and interventions across key areas such as hallucination reduction, reasoning improvement, and safety alignment. We hope this addresses the reviewer’s concern about the broader usefulness and impact of our work on understanding LLMs.
> > >
> > > **References:**
> > >
> > > [1] Abdaljalil et al. SAFE: A Sparse Autoencoder-Based Framework for Robust Query Enrichment and Hallucination Mitigation in LLMs. *arXiv preprint arXiv:2503.03032*.
> > >
> > > [2] Galichin et al. I Have Covered All the Bases Here: Interpreting Reasoning Features in Large Language Models via Sparse Autoencoders. In *CoRR*.
> > >
> > > [3] O’Brien et al. Steering Language Model Refusal with Sparse Autoencoders. In *ICML 2025 Workshop on Reliable and Responsible Foundation Models*.

---

> > > ### Author Response · Authors · 2025-08-02
> > > **Response to A1 and A2 (2/3)**
> > >
> > > > In computer vision, when the hype around XAI started researchers spend a lot of time to design more and new saliency (attribution) methods, sharpening the quality of generated importance maps. In the end, it turned out that the saliency methods that generated the most human-appealing images are the most useless in terms of debugging model behavior, meaning in doing something useful with them (see [4]).
> > >
> > > We thank the reviewer for highlighting important pitfalls in XAI research in computer vision, and we understand the concern that similar issues could arise in the context of LLMs. In response, we would like to elaborate on key differences between interpretability in vision and NLP, and explain why we believe XAI research in LLMs is less prone to the same pitfalls.
> > >
> > > In computer vision, explanations often take the form of saliency maps or concept activation heatmaps. [4] famously shows that many popular image saliency methods can produce nearly identical heatmaps even on randomized models. Therefore, visually pleasing heatmaps may reflect generic image structure more than the model’s true reasoning.
> > >
> > > LLMs present a different setting. Their “features” are abstract semantics rather than pixels. We can’t just black out a word in text without changing meaning; instead we clamp or remove latent features. Evaluation differs too: vision has ground-truth objects or textures to compare with, whereas in NLP we measure explanation quality by whether the text explanation predicts the feature’s activations. For instance, [5] finds that even GPT-4’s best explanations for GPT-2 neurons had high error rates and little causal power. This highlights that natural-language explanations (like those SAEs elicit) must be checked rigorously. In contrast, a poor image saliency can often be ruled out by a simple randomization test [4].
> > >
> > > In summary, vision XAI often focuses on image-based perturbations and visual heatmaps (with well-known pitfalls), while LLM XAI with SAEs focuses on latent semantic factors and textual probes. Our structured, component-based explanations and semantically similar negatives are tailored to these linguistic features: they aim to produce explanations that are faithful to the model’s combinatorial semantics, not just easy to read.
> > >
> > > **References:**
> > >
> > > [4] Adebayo et al. Sanity Checks for Saliency Maps. In *NeurIPS*.
> > >
> > > [5] Huang et al. Rigorously Assessing Natural Language Explanations of Neurons. In *EMNLP*.

---

> > > ### Author Response · Authors · 2025-08-02
> > > **Response to A1 and A2 (3/3)**
> > >
> > > > How does the method help in understanding LLMs **compared to Bills**? Can you demonstrate or summarize how the method/results improves our understanding of LLMs?
> > >
> > > While a full-scale comparison with [6] on downstream tasks is beyond the scope of this paper, we would like to use LLM refusal behavior as a concrete example to illustrate how the method of [6] can produce incorrect explanations, potentially hindering research, while our tree-based explainer provides more accurate and reliable interpretations.
> > >
> > > Understanding and controlling LLM refusal behavior in response to harmful prompts begins with identifying SAE features associated with refusal. For instance, on the Gemma 2 9B model, we queried the keyword “refusal” and found a feature whose explanation, generated by the method of [6], was:
> > >
> > > - negative words and phrases related to rejection or disapproval
> > >
> > > However, this explanation does not align with the actual activations of the feature, as shown below (with activated tokens in brackets):
> > >
> > > - 5-07[-0]9-77): An alternate arrangement composed of the Sasori, Oushi, Chameleon, Kajiki, and Super Shishi Voyagers.
> > > - Franco 2[0]16 approach. Finally, conclusions and future work are outlined in Section
> > > - 6-[0]7[-]09-11): An alternate arrangement composed of the Shishi, Hebitsukai, Chameleon, Kaijki, and Kuma Voyagers.
> > >
> > > This feature (layer 20, index 2403 in the GemmaScope-res-16K SAE) is viewable on Neuronpedia. The explanation from [6] incorrectly associates this feature with rejection, likely due to hallucination from the explainer. Such misinterpretations could mislead future work on refusal detection. In contrast, our tree-based explainer, which supports validation and refinement, yields the following, more accurate explanation:
> > >
> > > - The neuron is looking for specific numbers, particularly zero and numbers with high values such as 5 and 9, that may indicate a special case or an exceptional value in a document or data.
> > >
> > > We provide another example involving refusal. On Neuronpedia (Gemma 2 9B, layer 20, index 13138 of the GemmaScope-res-16K SAE), a feature is explained by [6] as:
> > >
> > > - instances of disbelief or doubt expressed through negation
> > >
> > > Yet its top activations are
> > >
> > > - I guess she doesn[’]t get the idea
> > > - He[’] a good dog
> > > - I[’]m so glad to
> > >
> > > Clearly, this explanation is misleading. The feature is not linked to semantic negation or disbelief, but rather to the presence of apostrophes. Our explainer correctly captures this:
> > >
> > > - The neuron is looking for the presence of an apostrophe character, typically used to indicate a contraction or possessive form, in a document.
> > >
> > > These examples highlight the importance of generating accurate explanations, especially for safety-critical behaviors like refusal. Our method not only avoids misleading outputs but also enables targeted and trustworthy interpretation of SAE features. As discussed earlier in our response, SAE-based explanations have been shown to be broadly useful for reducing hallucination, analyzing reasoning behavior, and improving safety in LLMs.
> > >
> > > We hope this discussion helps address the reviewer’s concerns about the practical utility and reliability of our approach. Please let us know if further clarification would be helpful.
> > >
> > > **References:**
> > >
> > > [6] Bills et al. Language models can explain neurons in language models. *OpenAI blog*.

---

> > ### Comment · Reviewer_i1y8 · 2025-08-01
> > **General response to the rebuttal**
> >
> > Thank you again for the rebuttal.
> >
> > 1. The reply to W4, Q4, Q5 is fine from my side and I hope that this will be improved in the paper.
> > 2. Related to Q6: In multiple answers in the rebuttals (also in replies to other reviewers) the authors mention case studies. Could the authors present the results of these case studies? For instance, to ensure that the LLM judge does a good job ("We do use an LLM judge to estimate feature complexity, but we design high-quality few-shot prompts and find that **its judgments align well with human intuition.**")
> > 3. Regarding Q1, sorry for mixing this.
> > 4. Q3: Can you please present the pseudo code?
> > 5. A2: Related to my earlier comment about A1. Can you demonstrate or summarize how the method/results improves our understanding of LLMs ("These methods are part of an iterative refinement process that significantly improves explanation accuracy and **helps us better understand LLMs.**"). Unfortunately, I still don't see this point albeit it is mentioned as key fact in the paper. To make my point clearer: I see that the generated explanations are improved over SOTA (I acknowledged this in my review). However, do we really learn something new by using these explanations? In computer vision, when the hype around XAI started researchers spend a lot of time to design more and new saliency (attribution) methods, sharpening the quality of generated importance maps. In the end, it turned out that the saliency methods that generated the most human-appealing images are the most useless in terms of debugging model behavior, meaning in doing something useful with them (see the paper "Sanity check of saliency methods" by Adebayo). Hence, the race to generate human-appealing (more accurate) explanations led us (more or less) nowhere. Hence, I urge the authors to demonstrate that the proposed method indeed increases our understanding of LLMs, for instance, by demonstrating that we can do something useful with it beyond the generation of human-appealing explanations and, hence, avoiding the pitfall from computer vision.

---

> > > ### Author Response · Authors · 2025-08-02
> > > **Response to Other Questions (1/2)**
> > >
> > > Thank you again for your prompt and constructive response. We address your remaining questions below.
> > >
> > > > Related to Q6: In multiple answers in the rebuttals (also in replies to other reviewers) the authors mention case studies. Could the authors present the results of these case studies? For instance, to ensure that the LLM judge does a good job ("We do use an LLM judge to estimate feature complexity, but we design high-quality few-shot prompts and find that **its judgments align well with human intuition.**")
> > >
> > > In our appendix and in responses to other reviewers, we provided case studies on the first three components of our contributions: semantically similar negatives, the tree-based explainer, and structured explanations. Here, we present additional case studies for the **complexity analyzer**, an LLM-based judge used to assess the complexity of SAE feature explanations. The few-shot examples used to prompt the LLM judge are detailed in Appendix A.3. Below, we present a sample of feature explanations along with their corresponding complexity scores to demonstrate alignment with human intuition.
> > >
> > > **Low complexity explanations:**
> > >
> > > - (complexity: 1.16) The word 'instrument' or 'instruments'.
> > > - (complexity: 0.81) The word 'risk'.
> > > - (complexity: 1.28) city
> > >
> > > These explanations are considered low in complexity because they correspond to specific tokens that activate the feature regardless of surrounding context.
> > >
> > > **Medium complexity explanations:**
> > >
> > > - (complexity: 3.00) The words 'different', 'differentiate', 'confused', or 'same' when in context of comparing two things.
> > > - (complexity: 3.38) Immunology-related terms, specifically 'antigen', and concepts related to immune response
> > > - (complexity: 3.25) The word 'become' and its variations in different contexts.
> > >
> > > These medium-complexity explanations typically correspond to a small set of tokens whose meaning depends on context. Accurately understanding these explanations requires recognizing variations of a word and interpreting them across different uses. This aligns with our few-shot examples in Appendix A.3, where context-dependent activations are considered more complex than simple token-level triggers. Nevertheless, these features still refer to relatively concrete concepts or actions and are not classified as highly complex.
> > >
> > > **High complexity explanations:**
> > >
> > > - (complexity: 4.25) emotions and psychological concepts such as fear, doubt, stress, anxiety
> > > - (complexity: 4.25) spiritual or emotional burdens and negative thoughts
> > > - (complexity: 4.13) social impact
> > >
> > > These high-complexity explanations involve abstract, high-level concepts that often require nuanced reasoning or interpretation. Empirically, we observe that such features tend to appear more frequently in the middle layers of the model, consistent with Figure 5 in our paper. These examples support the quality of the LLM judge’s assessments and align well with both our carefully designed prompting strategy and human intuition.
> > >
> > > In summary, we believe the results presented in Section 5.3 are reliable, as the complexity scores from the LLM judge generally reflect meaningful and interpretable distinctions across feature types.

---

> > > ### Author Response · Authors · 2025-08-02
> > > **Response to Other Questions (2/2)**
> > >
> > > > Q3: Can you please present the pseudo code?
> > >
> > > We apologize for omitting the pseudocode for the tree-based explainer in our previous response due to space limitations. We provide the complete procedure below.
> > >
> > > **Inputs:** Training and validation records for a single SAE feature, where each record contains:
> > >
> > > - The top 10 activating sentences
> > > - 10 complementary (non-activating) sentences
> > > - Ground-truth activations for each sentence
> > >
> > > **Output:** A natural language explanation for the given SAE feature
> > >
> > > **Algorithm:**
> > >
> > > 1. **Initialization**: Prompt the one-shot explainer (as in Bills et al.) independently $w$ times to generate $w$ initial explanations, which serve as the leaf nodes in the first iteration.
> > > 2. **Iterative Refinement** (for $i=1,2,\dots,d$):
> > >     1. **Evaluate current leaf nodes**:
> > >
> > >         For each explanation:
> > >
> > >         - Provide the explanation and validation sentences to a simulator LLM
> > >         - Predict SAE activations for the validation set using the explanation
> > >         - Compute a correlation score between predicted and ground-truth activations
> > >         - Construct a feedback message containing:
> > >             - The correlation score
> > >             - Simulated activations
> > >             - Ground-truth activations for the lowest-performing validation sentence
> > >     2. **Generate child explanations**:
> > >
> > >         For each current leaf node:
> > >
> > >         - Provide an LLM with its feedback message and the full conversation history from its ancestor nodes
> > >         - Use chain-of-thought prompting to generate $b$ improved explanations, which become new leaf nodes
> > >     3. **Evaluate new leaf nodes**:
> > >         - As in step (2.1), simulate and score the new explanations on the validation set
> > >         - Retain the top $w$ leaf nodes based on validation scores
> > >     4. **Early stopping**:
> > >         - If the highest validation score exceeds a predefined threshold, terminate early
> > >         - Otherwise, continue to the next iteration
> > > 3. **Final Output**:
> > >
> > >     Return the explanation with the highest validation score observed across all iterations.
> > >
> > >
> > > Note: Each iteration includes two evaluation phases on the validation set—one before generating child explanations (for feedback) and one afterward (for selection). Our implementation is available in the `TreeExplainer` class within `featureinterp/explainer.py` (included in the supplementary materials).
> > >
> > > We hope this clarifies our method and addresses your concerns. Please feel free to reach out with any further questions during the discussion period.
> > >
> > > > The reply to W4, Q4, Q5 is fine from my side and I hope that this will be improved in the paper.
> > >
> > > We thank Reviewer i1y8 for their thoughtful responses and valuable suggestions. These comments have greatly helped clarify the motivation and improve the overall quality of our paper. We will incorporate all related discussions into the revised version.

---

> > > > ### Comment · Reviewer_i1y8 · 2025-08-04
> > > >
> > > > Thank you once more for the detailed explanation.
> > > >
> > > > 1. I think we have different understandings about what a [case study](https://en.wikipedia.org/wiki/Case_study) is. I wouldn't call the manual analysis of a few results a case study but (maybe) a sanity check (hopefully, not cherry-picked!). A case study would require more scientific rigor.
> > > > 2. Thank you for posting the pseudo-code, which makes it clearer.
> > > > 3. Regarding that SAEs improve our understanding. I agree with the authors with respect to the described highlights where SAEs helped in understanding LLMs. However, if it is claimed in the paper that the proposed method *improves the understanding of LLMs*, shouldn't the paper present a study along these lines to really show that?
> > > >
> > > > Looking forward to hearing the authors thoughts and thank you for your time in advance.

---

> > > > > ### Author Response · Authors · 2025-08-04
> > > > > **Response to Reviewer i1y8**
> > > > >
> > > > > We thank Reviewer i1y8 once again for their thoughtful engagement and constructive feedback, which has significantly helped us improve our paper. Please find our responses below:
> > > > >
> > > > > **On the use of the term “case study”:** We agree with the reviewer that our example analysis does not meet the standards of a formal case study, which typically requires stronger methodological rigor. Our intent was to present a sanity check showing that the LLM judge’s complexity assessments broadly align with human judgment. To mitigate concerns about cherry-picking, we’ve included the full list of complexity scores for the first 10 features of layer 12 of Gemma at the end of this response. Upon inspection, these scores appear reasonable, and manual assessment would likely yield similar results. However, manually annotating the entire experimental dataset is prohibitively expensive, which is why we rely on the LLM analyzer and validate it through these checks. We will clarify the terminology and framing in the final paper to avoid overstating this component.
> > > > >
> > > > > **On the claim that our method improves understanding of LLMs:** We appreciate the reviewer’s question regarding the strength of this claim. We believe the claim is supported by the following points:
> > > > > 1. **Empirical insights enabled by our method:** Section 5.3 is devoted to analyzing how complexity and polysemanticity vary across layers and model sizes. For example, we show that both measures tend to peak in the middle layers and increase with model ability—patterns that were not quantitatively studied in prior work. These findings offer new perspectives on how LLMs structure internal representations and thus contribute to a deeper understanding of their behavior.
> > > > > 2. **Improved explanations contribute to better interpretability:** Since SAEs are widely used for interpreting LLM internals, improvements in explanation quality directly enhance our ability to understand neuron functions and the circuits they participate in. This also supports practical downstream tasks like detecting hallucinations and debugging models.
> > > > > 3. **Understanding internal mechanisms is itself valuable:** We believe that advancing understanding of low-level mechanisms is a legitimate and valuable form of interpretability research. For example, Bills et al. make a similar point in their SAE methodology paper:
> > > > >
> > > > >    > “We believe our methods could begin contributing to understanding the high-level picture of what is going on inside transformer language models.”
> > > > >
> > > > >    Like theirs, our method focuses on improving explanation quality, and we similarly believe this is a critical step toward broader understanding.
> > > > >
> > > > > That said, we recognize the reviewer’s concern that the current framing of this claim may be too broad or strong. We are open to softening the statement in the final version—for example, by specifying that we provide **a first step** toward understanding certain behaviors or **a clearer view** of internal mechanisms. We believe our contributions remain valuable and impactful even without this claim, particularly given the strong improvements over prior work in explanation quality and the method’s applicability to real-world interpretability tasks.
> > > > >
> > > > > We sincerely thank the reviewer again for their feedback and consideration. Please let us know if further clarification would be helpful.
> > > > >
> > > > > **Appendix:**
> > > > > A full list of explanation complexities for the first 10 features of layer 12 of Gemma:
> > > > > ```
> > > > > [3.25] The word 'become' and its variations in different contexts.
> > > > > [2.36] Common words or phrases indicating change of state, such as 'becomes'.
> > > > > [3.06] The use of intensifying adverbs.
> > > > > [1.27] Wright
> > > > > [1.15] netto
> > > > > [1.17] sie
> > > > > [1.09] Webber
> > > > > [1.02] Cliff Richard
> > > > > [1.13] born
> > > > > [1.73] ( date of birth )
> > > > > [3.53] Mathematical expressions and formulas, particularly those containing symbols like $, \, and $.
> > > > > [2.95] The character 'lambda' and related scientific concepts.
> > > > > [2.23] Physical quantities such as length, radius, and size.
> > > > > [2.47] The word 'contact' in various contexts, including forms, emails, and contact pages.
> > > > > [1.73] Words related to forms, such as 'form' and 'forms'.
> > > > > [3.00] The words 'different', 'differentiate', 'confused', or 'same' when in context of comparing two things.
> > > > > [1.92] Specific biological terms like 'lymphocyte', 'B cell', 'antigen', or 'immune functions'
> > > > > [1.80] Chemical symbols or molecular terms, such as 'beta', 'g', or 'F'
> > > > > [1.74] Product codes or names with specific formatting, such as 'Y', or 'OHN'
> > > > > [3.38] Immunology-related terms, specifically 'antigen', and concepts related to immune response
> > > > > [2.03] Cell-related terms, like 'B cell', 'lymphocyte' or 'immune functions'
> > > > > [1.83] The word 'year', especially when used in a temporal context (e.g., 'this year', 'per year')
> > > > > [3.11] The phrase 'year' with adjacent positive or optimistic words
> > > > > [1.42] The word 'season'
> > > > > [1.27] free King The Kid ringtones
> > > > > [1.20] DU Battery Saver
> > > > > [1.86] World Conqueror
> > > > > [1.31] VPN
> > > > > [1.83] Google Speed Tracer
> > > > > [1.27] The word 'want'
> > > > > ```

---

> > > > > > ### Comment · Reviewer_i1y8 · 2025-08-05
> > > > > >
> > > > > > I appreciate the authors response and the entire rebuttal. I will increase my score to **borderline accept** assuming that the authors will apply the promised changed.
> > > > > >
> > > > > > During the rebuttal, I have read the reviews of the other reviewers that evaluate the contribution (the improvement of Bills et al. approach) as significant and valuable. Hence, I have reconsidered my assessment and agree that this an interesting contribution for the community.
> > > > > >
> > > > > > My change in the assessment is also based on the promises of the authors to change the paper as discussed during the rebuttal (e.g., presentation of pseudo code). Moreover, I urge the authors to realistically state their contributions and to not oversell it unrealistically (e.g., overselling of "understanding" or "case study"). A realistic reflection of the contributions and limitations is appreciated by the community and the conference to know in advance what to expect and where are points for improvement. Maybe the usage of the limitations section is a good place for that.
> > > > > >
> > > > > > Once more, I thank the authors for the good rebuttal and wish all the best.

---

> > > > > > > ### Author Response · Authors · 2025-08-05
> > > > > > > **Response to Reviewer i1y8**
> > > > > > >
> > > > > > > We sincerely thank the reviewer for their thoughtful engagement throughout the review process and for the constructive suggestions. We appreciate your reconsideration of the paper and are glad that the contribution is seen as valuable to the community.
> > > > > > >
> > > > > > > We fully agree on the importance of presenting our contributions with clarity and realism. As suggested, we will revise the paper to incorporate all the discussed changes, including the addition of pseudocode and a more measured articulation of our claims. We will also make use of the limitations section to explicitly reflect on the scope and boundaries of our work.
> > > > > > >
> > > > > > > Thank you again for your time and support.

---

### Official Review · Reviewer_gHHG · 2025-07-04

**Clarity:** 2
**Significance:** 3
**Originality:** 3
**Rating:** 5
**Confidence:** 3

**Summary:**

This paper addresses the problem of explaining features extracted by sparse autoencoders (SAEs) --- a method to obtain disentangled representations for a neuron in models such as LLMs --- via text in a faithful manner. A common way to do this is to look at what activates a given SAE latent and then ask an LLM for an explanation that encompasses highly activating inputs and excludes low activating inputs. Simulation experiments are then performed to check if predicted activations based on just the input and the explanation match true activations. This work recognizes that getting such an explanation from an LLM is challenging, since the explanation may be overly broad, which would result in high recall but low precision of simulation. It proposes using structured explanations and a tree based search to obtain better explanation candidates, and similarity based retrieval methods to better retrieve negative examples, i.e. examples that do not activate the latent. Experiments are performed on Pile with three LLMs to show that the proposed approach yields more false positives (i.e. avoids having too broad explanations), has better scores on simulation experiments, and can be used to analyze explanation complexity and polysemanticity.

**Questions:**

Please refer to the Weaknesses. I would be happy to increase my score if the concerns on motivation and having a more granular analysis are addressed.

**Ethical Concerns:**

["NO or VERY MINOR ethics concerns only"]

**Final Justification:**

I appreciate the examples provided, including negative examples. It would be very helpful to include several of them with a detailed discussion in the revision, as this makes the paper much stronger.

Overall, my major concerns have been addressed, so I would be happy to increase my score to 5.

An additional question about feature splitting: the false positive correlation is interesting, but is just a proxy metric. What would be much more interesting is to see if these structured explanations can reveal that feature splitting happens, i.e. if you list out the explanations in the tree, does it show the exceptions that are not covered by the latent due to feature splitting? This in my opinion would be a much more interesting analysis.

**Limitations:**

Yes, but an additional limitation to consider is the reliance on LLMs for obtaining explanations.

**Paper Formatting Concerns:**

Yes, the formatting used has a non-standard font that is different from the NeurIPS Latex template.

**Quality:**

3

**Strengths And Weaknesses:**

## Strengths

1. The paper addresses an important problem, since getting faithful explanations is important when using models in safety critical domains.
2. The proposed approach is simple but principled, and quantitative results appear to show that it is effective.

## Weaknesses

1. The key weakness for me is that the paper lacks qualitative examples that show why the proposed method is useful. While quantitatively one can see that it is better at finding false positives (Figure 3) and getting better scores (Figure 4), how does this finally affect the kind of explanations that are obtained for SAE latents? For instance, are more granular explanations obtained? Showing examples of explanations for latents along with what inputs activate them would be very helpful (including negative examples). Right now it is not very clear how useful the approach would be in practice, and whether the quantitative improvements are significant. The examples shown in the Appendix seem to primarily be prompting examples, but if this is provided there please point to them in the rebuttal.

2. A problem that has been observed with SAEs is feature absorption and feature splitting (e.g. discussed in [1]). Intuitively, one would expect that this approach would be able to detect such instances better. This could be a very useful evaluation to have to show the practical benefits of the approach, since e.g. situations involving feature splitting are likely to show up false positives as discussed in the paper. It would also help understand if the sentence encoder similarities used in the paper are helpful for finding ideal candidate sentences. Less important, but a study on different SAE types (e.g. using Matryoshka SAEs) could also be useful.

3. The trends shown for explanation complexity and polysemanticity (Figure 5) appear somewhat weak, and a more granular analysis would be helpful. For example, the results also depend on how good the trained SAE is, which would vary across models and layers (in terms of reconstruction loss, latent dimension size, sparsity, etc.) and could affect the conclusions made. Additionally, similar to Weakness 1, having examples would also help here, particularly for polysemanticity. For example, are the explanations for the polysemantic neurons conceptually similar or very distinct?

---

> ### Author Rebuttal · Authors · 2025-07-31
>
> We thank Reviewer gHHG for their thoughtful review. Below are our responses to their concerns.
>
> ---
>
> **W1.** The key weakness for me is that the paper lacks qualitative examples that show why the proposed method is useful. While quantitatively one can see that it is better at finding false positives (Figure 3) and getting better scores (Figure 4), how does this finally affect the kind of explanations that are obtained for SAE latents? For instance, are more granular explanations obtained? Showing examples of explanations for latents along with what inputs activate them would be very helpful (including negative examples). […]
>
> **A1.** Thank you for this constructive suggestion. The tree-based explainer can generate more granular, accurate, and concise explanations compared to the one-shot explainer, thanks to its ability to iteratively refine explanations based on evaluation feedback. We demonstrate this through three examples where the tree-based explainer outperforms the one-shot explainer, plus one negative example showing its limitations.
>
> **Example 1:** A birthdate feature
>
> - One-shot explanation components (score: 0.062):
>     - Names of people, specifically first and last names.
>     - Date formats and structures, including days, months, and years.
> - Tree-based explanation components (score: 0.621):
>     - born
>     - date of birth in parentheses
> - Example activations (with activated tokens in bold):
>     - Anabel Englund (**born September** 1 **,** 1**9**92 **) is** an American singer and songwriter
>     - Kim Soo-yong (**born September 23, 19**2**9) is** a South Korean film director
>     - Simon **Blair Doull (born 6 August 19**6**9) is** a New Zealand radio personality, commentator and former international cricketer
>
> This feature activates on birthdates of people in parentheses. While the one-shot explanation broadly includes names and date formats, the tree-based explanation captures the specific pattern more accurately.
>
> **Example 2:** An import and JavaScript keywords feature
>
> - One-shot explanation components (score: -0.025):
>     - Programming syntax related to imports, specifically import statements.
>     - Code comments.
>     - Specific programming keywords.
> - Tree-based explanation components (score: 0.117):
>     - JavaScript/React keywords and syntax
>     - import and export statements
>     - slider and view related terms
>     - querySelector and document
>     - formData and FormData
> - Example activations:
>     - **import {** NUM_BINS, PRES**ETS** } **from "./** config **";**
>     - **export default** Object.**freeze**({
>     - **const grid = document.querySelector('#** Symbols **.grid');**
>
> This polysemantic feature activates on various programming contexts, including import statements and JavaScript keywords. The tree-based explanation provides more detailed coverage of these activation patterns.
>
> **Example 3:** A parenthesis feature
>
> - One-shot explanation components (score: 0.100):
>     - Mathematical and programming syntax, especially with context of abstract algebra and mathematical concepts.
>     - Use of parenthesis '('
> - Tree-based explanation components (score: 0.602):
>     - '('
>     - ')'
> - Example activations:
>     - create a **(** mutable **)** object first, and at certain scenario I need to make it immutable
>     - I allow other (mac **)** machines to access an IIS7 hosted MVC3 application on my (win7 **)** machine
>     - I'm looking for a formula to calculate the **(** product **?)** of an arithmetic series
>
> This feature simply activates on parentheses. The tree-based explanation captures this concisely, while the one-shot explanation includes unnecessary context about mathematical concepts.
>
> **Negative example:** A people name feature
>
> - One-shot explanation components (score: 0.251):
>     - Names of people, including first and last names.
>     - Proper nouns, including names of songs, albums, movies, and places.
> - Tree-based explanation components (score: 0.039):
>     - Wright
>     - netto
>     - King
>     - Cliff Richard
>     - Lloyd Webber
> - Example activations:
>     - billionaire Nirav **Modi**, could have cost $2 billion
>     - 1965 popular song with lyrics by Hal **David** and music composed by Burt Bach**arach**
>     - "You have seen into the heart of my music," **Andrew Lloyd Webber** said to Gale **Edwards** in a trans
>
> This feature activates on people's names, but the tree-based explainer overfits by producing a list of specific names. Such cases are rare in our experiments.
>
> ---
>
> **W2.** A problem that has been observed with SAEs is feature absorption and feature splitting. Intuitively, one would expect that this approach would be able to detect such instances better. This could be a very useful evaluation to have to show the practical benefits of the approach, […]
>
> **A2.** Thank you for this helpful suggestion. We hypothesized that feature splitting and absorption would result in more false positives, and tested this empirically. Following [1], we examined SAE features that activate on words starting with specific English letters. We measured feature splitting and false positives from the simulator (which activates on words beginning with particular letters). Here are the correlation scores at different feature splitting thresholds:
>
> | F1 threshold | Correlation coefficient |
> | --- | --- |
> | 0.03 | 0.553 |
> | 0.02 | 0.578 |
> | 0.01 | 0.637 |
>
> The high correlation ($r>0.5$) between feature splitting and false positive rate suggests our approach offers an efficient way to detect feature splitting without training multiple k-sparse probes.
>
> We also found a correlation of 0.1042 between feature absorption and false positives—statistically significant but more subtle than feature splitting.
>
> **References:**
>
> [1] Chanin et al. A is for Absorption: Studying Feature Splitting and Absorption in Sparse Autoencoders. *arXiv preprint arXiv:2409.14507*.
>
> ---
>
> **W3.** The trends shown for explanation complexity and polysemanticity appear somewhat weak, and a more granular analysis would be helpful. For example, the results also depend on how good the trained SAE is, which would vary across models and layers (in terms of reconstruction loss, latent dimension size, sparsity, etc.) and could affect the conclusions made.
>
> **A3.** You raise an important point about how SAE training quality affects feature complexity and polysemanticity. While we kept latent dimensions consistent across layers, reconstruction loss and sparsity varied. For a more detailed analysis, we measured correlations between complexity and reconstruction loss ($r_1$), and between polysemanticity and reconstruction loss ($r_2$), across different layers:
>
> | Model | $r_1$ | $r_2$ |
> | --- | --- | --- |
> | Gemma | 0.320 | 0.293 |
> | Llama | 0.520 | 0.461 |
> | GPT-2 | 0.671 | 0.151 |
>
> The positive correlation between these metrics and reconstruction loss makes sense—more complex and polysemantic features are inherently harder for SAEs to learn.
>
> We also examined correlations between complexity and L0 ($r_3$), and between polysemanticity and L0 ($r_4$):
>
> | Model | $r_3$ | $r_4$ |
> | --- | --- | --- |
> | Gemma | 0.262 | -0.093 |
> | GPT-2 | 0.229 | 0.086 |
>
> Llama was excluded as it uses top-k activation. The results show complexity correlates positively with L0, while polysemanticity shows no significant correlation.
>
> ---
>
> **W4.** Additionally, similar to W1, having examples would also help here, particularly for polysemanticity. For example, are the explanations for the polysemantic neurons conceptually similar or very distinct?
>
> **A4.** For polysemantic neurons, structured explanations better capture conceptually distinct components compared to unstructured explanations. Here are two illustrative examples:
>
> **Example 1:**
>
> - Unstructured explanation (score: 0.894): The neuron is looking for mentions of specific types of devices, tools, or equipment, particularly when referred to by their common names or technical terms, such as "instruments", "instrument", or warning messages like "warnings" or "Warning".
> - Structured explanation (score: 0.967):
>     - The word 'instrument' and its plural 'instruments'.
>     - Warnings in code, such as 'Warning' and 'warnings'.
> - Example activations:
>     - Copyright (C) 2004 Texas **Instruments**
>     - This is a follow up to Is -Wreturn-std-move clang **warning** correct in case of objects in the same hierarchy
>
> This polysemantic feature activates on both "instruments" and "warning"—distinct concepts that structured explanations capture more clearly.
>
> **Example 2:**
>
> - Unstructured explanation (score: 0.493): The neuron is looking for phrases or sentences that express sincerity, emphasis, or encouragement, often marked by words like "mean", "seriously", "come on", or phrases that start with "let's be honest". It also seems to activate on certain punctuation marks like commas that precede or follow these phrases.
> - Structured explanation (score: 0.589):
>     - The word 'on' in various contexts.
>     - The word 'mean' when used for emphasis.
>     - Expressions of informality or emphasis like 'come on', 'hey', or 'seriously'.
>     - Commas in certain phrases.
> - Example activations:
>     - Hang **on** for a minute...we're trying to find some more stories you might like
>     - By other js files, I **mean** the files you include in popup.html
>     - Come **on** in and stay for the video
>
> This feature also activates on conceptually distinct patterns, which structured explanations capture more effectively.
>
> ---
>
> We'll incorporate these additional results and discussion in our paper. We appreciate your feedback and are happy to address any further questions.

---

> > ### Comment · Reviewer_gHHG · 2025-08-08
> >
> > Thank you for the detailed response! I appreciate the examples provided, including negative examples. It would be very helpful to include several of them with a detailed discussion in the revision, as this makes the paper much stronger.
> >
> > Overall, my major concerns have been addressed, so I would be happy to increase my score to 5.
> >
> > An additional question about feature splitting: the false positive correlation is interesting, but is just a proxy metric. What would be much more interesting is to see if these structured explanations can reveal that feature splitting happens, i.e. if you list out the explanations in the tree, does it show the exceptions that are not covered by the latent due to feature splitting? This in my opinion would be a much more interesting analysis.

---

> > > ### Author Response · Authors · 2025-08-08
> > > **Response to Reviewer gHHG**
> > >
> > > We sincerely thank the reviewer for their positive feedback and for raising valuable suggestions to strengthen our work. We will incorporate the positive and negative examples, along with detailed discussions, into the revision as suggested.
> > >
> > > Regarding the question on feature splitting, we agree that this is a very interesting and promising direction for analysis. Simply listing structured explanations is likely insufficient to reveal such cases in full. Instead, we would need to leverage the tree-based explainer’s ability to iteratively reflect and refine explanations using false positive samples as prompts. This would allow us to surface exceptions and boundaries missed by the latent feature. We plan to explore this approach in additional experiments, which will require a more comprehensive setup than our current evaluation.
> > >
> > > We greatly appreciate the reviewer’s insights, which will help us further improve the clarity and depth of the paper.

---

> ### Author Response · Authors · 2025-08-06
> **Follow-up on Our Rebuttal**
>
> Dear Reviewer gHHG,
>
> Thank you again for your valuable comments and detailed review of our paper. We have carefully considered your feedback and submitted a comprehensive rebuttal addressing all the points you raised.
>
> As the discussion phase is approaching its end, we would like to kindly confirm whether we have adequately addressed your concerns. If any questions remain or further clarification is needed, please don’t hesitate to let us know—we are happy to respond promptly.
>
> If you feel that our responses sufficiently address your concerns, we would be grateful for your consideration in revisiting your evaluation.
>
> We sincerely look forward to your feedback.
>
> Best regards,
>
> The Authors

---

### Official Review · Reviewer_q24V · 2025-07-10

**Clarity:** 3
**Significance:** 3
**Originality:** 3
**Rating:** 5
**Confidence:** 4

**Summary:**

This paper proposes three major innovations to the standard automated interpretability pipeline for sparse autoencoder features. First, it introduces a similarity-based strategy to find semantically related features to a given query feature. The LLM explainer is then asked to generate explanations that make it possible to distinguish activating examples for the query feature from activating examples for the semantically related features. Second, it introduces a JSON-based format for structured explanations, enabling the use of more sophisticated explanations than were used in prior work. Third, it introduces a tree-based approach for iteratively improving structured explanations.

**Questions:**

1. Have you tested your innovations on lower quantiles of the activation distribution? Do you anticipate that your results will be qualitatively different on, say, the middle tercile of the feature activations?

**Ethical Concerns:**

["NO or VERY MINOR ethics concerns only"]

**Limitations:**

yes

**Quality:**

3

**Strengths And Weaknesses:**

Strengths:
- Structured explanations enable a more direct measurement of the degree of polysemanticity in a feature (number of explanation components)
-  Tree-based explanation is novel and appears to open up new directions for research

Weaknesses:
- Focuses on top-activating examples, which are known to be significantly more interpretable than medium or low activation quantiles
- Tree-based explanation method is computationally intensive

---

> ### Author Rebuttal · Authors · 2025-07-31
>
> We thank Reviewer q24V for the constructive review. We address your concerns as follows.
>
> ---
>
> **W1.** Focuses on top-activating examples, which are known to be significantly more interpretable than medium or low activation quantiles.
>
> **A1.** Thank you for this observation. We focus on the top-activating quantile because SAE features only activate on a tiny fraction of tokens, making sentences in medium and low activation quantiles essentially random. Our evaluation approach on top-activating quantiles aligns with standard practices [1].
>
> To verify that our methods remain superior across different activation levels, we tested on the 10th activation quantile (out of 1000 total). For lower quantiles, sentences would be uninterpretable due to the sparse nature of SAE activations. In this experiment, we combined sentences from this quantile with an equal number of random sentences to form training sets for generating explanations and test sets for evaluation. This setup mirrors our paper and [1], with the only difference being that we replaced top-activating sentences with those from another quantile.
>
> Results for the 10th quantile are shown below:
>
> | Method | Layer 10 | Layer 20 |
> | --- | --- | --- |
> | One-shot, unstructured | 0.099 | 0.118 |
> | One-shot, structured | 0.172 | 0.107 |
> | Tree, unstructured | 0.121 | 0.144 |
> | Tree, structured | 0.222 | 0.165 |
>
> These results demonstrate that structured explanations consistently outperform unstructured ones, and the tree explainer generally surpasses the one-shot explainer. However, we should note that sentences in lower quantiles tend to be more random, making scores from those quantiles less informative.
>
> **References:**
>
> [1] Bills et al. Language models can explain neurons in language models. *OpenAI blog*.
>
> ---
>
> **W2.** Tree-based explanation method is computationally intensive.
>
> **A2.** We agree that the tree-based explainer requires more computational resources than the one-shot explainer, particularly at larger depths and widths. However, this cost comes with a significant performance improvement for SAE feature explanations (~20% increase in correlation score). This represents a substantial advance in mechanistic interpretability approaches and enhances our understanding of LLMs. Given that SAEs are widely used for understanding and controlling LLM behaviors, accurate feature explanations are essential. Additionally, the tree-based explainer's depths and widths can be adjusted to balance efficiency and accuracy.
>
> ---
>
> **Q1.** Have you tested your innovations on lower quantiles of the activation distribution? Do you anticipate that your results will be qualitatively different on, say, the middle tercile of the feature activations?
>
> **A3.** As shown in **A1**, we tested our methods on the 10th quantile and consistently observed improvements over baselines. This confirms our approaches' superiority across different activation quantiles. For even lower quantiles, SAE activations are essentially zero, rendering the sentences uninterpretable.
>
> ---
>
> We hope our response addresses your concerns. If you have further questions, we will be happy to address them during the discussion period.

---

> ### Author Response · Authors · 2025-08-06
> **Follow-up on Our Rebuttal**
>
> Dear Reviewer q24V,
>
> Thank you once again for your thoughtful and constructive feedback on our submission. We deeply appreciate your engagement during the review process and are grateful for your positive assessment of our work.
>
> As the discussion phase is drawing to a close, we wanted to kindly confirm whether there are any remaining concerns or clarifications we can provide. We are happy to respond to any further questions you may have.
>
> Thank you again for your time and support.
>
> Best regards,
>
> The Authors

---

### Note · Authors · 2025-08-12

Dear PCs, SACs, ACs, and reviewers,

We sincerely thank you for the time and effort you dedicated to reviewing our paper and for providing valuable feedback that has helped us strengthen our work. We are glad to have addressed many of your concerns and to have received both positive feedback and constructive suggestions. As promised, we summarize below the changes we plan to incorporate into the revised version of our paper:
- **Reviewer q24V**: We will add discussions on results for other activation quantiles, where our methods consistently outperform the baselines.
- **Reviewer gHHG**: We will include a more granular analysis of feature complexity and polysemanticity, provide detailed discussions of positive and negative examples of generated explanations, and elaborate on the correlation with feature splitting.
- **Reviewer i1y8**: We will incorporate our discussion points on motivation, design choices, pseudo code for the tree-based explainer, and other clarifications. We will also revise our contribution statements to avoid overselling and add a more realistic reflection of our contributions in the limitations section.
- **Reviewer LBic**: We will add discussions on our extensive sanity checks and other measures to confirm alignment with human evaluations, justify our dataset choice, and clarify the polysemantic nature of SAE features in the latent space.
- **All reviewers**: We will address the suggested formatting improvements, correct typos, and add missing citations.

Finally, we deeply appreciate the thoughtful reviews, as well as the efforts of the PCs, SACs, and ACs in facilitating this process!

---

### Decision · Program_Chairs · 2025-09-17

**Decision:**

Accept (poster)

**Comment:**

This paper proposes a few simple modifications of standard automated interpretability pipelines in sparse autoencoder features, including the tree-based structure that helps to extract more precise explanations and allows better tackling of the polysemanticity. The reviewers are positive after rebuttal, where the authors provided additional qualitative examples, ablation studies, and clarifications between prior work, as well as additional applicability with related literature. The authors are also committed to the promised revisions (in the author final remark) to further strengthen the manuscript. Thus, an acceptance is recommended.